# FIXED NON-NEGATIVE ORTHOGONAL CLASSIFIER: INDUCING ZERO-MEAN NEURAL COLLAPSE WITH FEATURE DIMENSION SEPARATION

**Hoyong Kim, Kangil Kim**[*]
AI Graduate School, GIST[†], Republic of Korea
hoyong.kim.21@gm.gist.ac.kr, kangil.kim.01@gmail.com

## ABSTRACT

Fixed classifiers in neural networks for classification problems have demonstrated cost efficiency and even outperformed learnable classifiers in some popular benchmarks when incorporating orthogonality (Pernici et al., 2021a). Despite these advantages, prior research has yet to investigate the training dynamics of fixed classifiers on neural collapse. Ensuring this phenomenon is critical for obtaining global optimality in a layer-peeled model, potentially leading to enhanced performance in practice. However, the neural collapse cannot explain the collapse phenomenon in the fixed classifier when its shape is not a simplex ETF. To overcome the limits, we exploit additional constraints to the layer-peeled model: non-negativity and orthogonality. Then, we propose a *fixed non-negative orthogonal classifier*, which makes a layer-peeled model with the fixed classifier have the global optimality and the max-margin in decision by inducing *zero-mean neural collapse*. Building on this foundation, we exploit a *feature dimension separation* inherent in our classifier for further purposes: (1) enhances softmax masking by mitigating feature interference in continual learning and (2) tackles the limitations of mixup on the hypersphere in imbalanced learning. We conducted comprehensive experiments on various datasets and demonstrated significant performance improvements.

## 1 INTRODUCTION

Classification models built on neural networks have an encoder that generates features from input samples and a classifier in the datasets. Many works have focused on the classifier and one of them has fixed the weight matrix in the classifier, called a *fixed classifier*. The fixed classifier has adopted not only orthogonal and Hadamard matrices (Hoffer et al., 2018), but also geometric shapes, including simplex (Pucci et al., 2021; Mettes et al., 2019), cube, and orthoplex (Pernici et al., 2019; 2021a), exhibiting cost efficiency and comparable performance. Even if the fixed classifier performed better in the presence of orthogonality (Pernici et al., 2021a), it has not been deeply studied in terms of neural collapse (Papyan et al., 2020), a recently clarified phenomenon for training classification models.

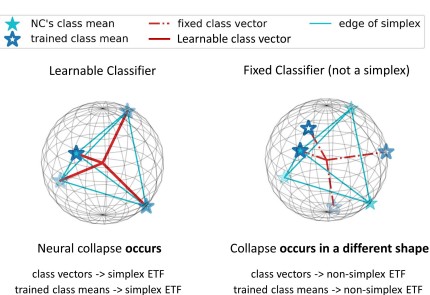

Figure 1: Motivation: *How does the collapse between class means and class weight vectors occur in the fixed classifier when its shape is not a simplex ETF?*

In the stream of neural collapse, prior researchers analyzed the cause of the phenomenon on a *layer-peeled model* (LPM), and proved that the neural collapse emerges when the LPM has the global optimality with various constraints (Ji et al., 2022; Han et al., 2021; Lu & Steinerberger, 2020; Mixon et al., 2020; Kothapalli et al., 2022; Yang et al., 2022b) by inducing the collapse of class means and class weight vectors onto mutually equiangular and equidistant vertices, referred to *simplex equiangular tight frame* (simplex ETF). Previous research has shown the effectiveness of neural

---

[*]: corresponding author, [†]: Gwangju Institute of Science and Technology

collapse properties by anchoring classifiers as a simplex ETF (Yang et al., 2022b; 2023a). However, the collapse does not occur at a simplex ETF in the fixed classifier due to their geometrical limitations when the shape of weight matrix in it is not a simplex ETF, as shown in Figure 1.

In this paper, we propose a *zero-mean neural collapse* to explain the collapse phenomenon of classification models with fixed orthogonal classifiers in non-negative Euclidean space by centering class means to the origin, not their global mean. Then, we demonstrate that non-negativity and orthogonality are valuable for LPM with fixed classifiers, allowing for the global optimality and max-margin in decision simultaneously. Finally, we propose a *Fixed Non-negative Orthogonal* (FNO) classifier and prove that zero-mean neural collapse occurs when LPM with an FNO classifier has the global optimality while inducing the max-margin in decision. In addition, the FNO classifier becomes linearly independent due to non-negativity and orthogonality. As a result, some elements in the last-layer feature engaged to one class will not be able to affect other classes. This *feature dimension separation* (FDS) synergizes with masked softmax by decreasing interferences between classes in continual learning and enables *arc-mixup* by adjusting the mixup strategy on the hypersphere to work correctly in imbalanced learning.

Formally, our concrete contributions are as follows:

- **Zero-mean Neural Collapse** (section 4) We propose a *zero-mean neural collapse* to analyze the collapse phenomenon in training classification model with the fixed orthogonal classifier.

- **Fixed Non-negative Orthogonal Classifier** (section 5) We propose a *fixed non-negative orthogonal classifier* and prove its theoretical benefits in orthogonal layer-peeled model with zero-mean neural collapse.

- **Benefits of Feature Dimension Separation** (section 6, 7) We demonstrate the impacts of our methods with masked softmax in continual learning and arc-mixup in imbalanced learning.

## 2  RELATED WORK

**Fixed Classifier, Neural collapse, and Orthogonality.**    Fixed simplex ETF classifiers have been proposed (Liang & Davis, 2023) and have demonstrated superior performance in continual learning (Yang et al., 2023a) and imbalanced learning (Thrampoulidis et al., 2022; Yang et al., 2022b). In contrast, although orthogonality has also shown advantages in various environments (Appendix A.2), it has not been deeply studied about the collapse phenomenon in training classification models with the fixed orthogonal classifier. Likewise, the fixed orthogonal classifier has not been utilized in continual learning and imbalanced learning although they have demonstrated the effectiveness in a similar way to the simplex ETF (Hoffer et al., 2018; Pernici et al., 2019; 2021a;b). To the best of our knowledge, the reason for this is that enforcing orthogonality results in the classifier being orthogonal to each other while class means centering to their global mean converge to a simplex ETF. A few research have analyzed neural collapse with orthogonality by using LPM with positive features (Nguyen et al., 2022) and exploiting unconstrained features model (Tirer & Bruna, 2022), but they did not freeze the classifier and did not consider non-negativity and orthogonality in the classifier simultaneously. RBL with an orthogonal-constrained classifier (Peifeng et al., 2023) has also shown the effectiveness of orthogonality in imbalanced learning. However, their classifier was not analyzed with LPM and did not take into account non-negativity, which is an important factor that causes feature dimension separation in the weight matrix of the fixed orthogonal classifier. The findings of neural collapse under MSE loss (Zhou et al., 2022a) yield similar results to ours. Specifically, the class means and class weight vectors collapse in orthogonal shape. However, their classifier was not fixed and the cross entropy loss, which is the most widely used loss function in classification models, was not utilized. These differences highlight the improved utility of our work.

Based on the different geometric feature of orthogonality, we have developed methods with the conviction that a fixed orthogonal classifier has potential in both continual learning and imbalanced learning much like a fixed simplex ETF, despite it not converging to a simplex ETF. With the intuition, we examined the collapse phenomenon that occurs when training a classification model with the fixed orthogonal classifier in the non-negative Euclidean space, which we have termed *zero-mean neural collapse*. We then proposed a *fixed non-negative orthogonal classifier* that satisfies the properties of zero-mean neural collapse while also providing benefits in continual learning and imbalanced learning by class-wise separation of the last-layer feature dimensions.

## 3 PRELIMINARIES

**Notations.** Let $\mathbb{D} = \{(\boldsymbol{x}_i, c_i) \mid 1 \leq i \leq N\}$ denotes a dataset which consists of $N$ input samples $\boldsymbol{x}$ and their class labels $c$. $\boldsymbol{y}_{c_i} \in \mathbb{R}^K$ is a one-hot vector whose index value of 1 corresponds to the class label of the $i$-th input sample among $K$ classes. We define the encoder of the classification model as a function $f_\theta$ which is parameterized by $\theta$, $f : \mathbb{R}^{dim(\boldsymbol{x})} \to \mathbb{R}^D$. Let $\mathbf{W} \in \mathbb{R}^{D \times K}$ and $\mathbf{b} \in \mathbb{R}^K$ are weight and bias parameters in the classification layer. Then, the last-layer feature $\mathbf{h} \in \mathbb{R}^D$ and its probability vector $\mathbf{p} \in \mathbb{R}^K$ are defined as below:

$$
\begin{aligned}
\mathbf{h} &= \text{RELU}\left(f_\theta\left(\boldsymbol{x}\right)\right) \\
\mathbf{p} &= \text{SOFTMAX}\left(\mathbf{W}^\intercal \mathbf{h} + \mathbf{b}\right).
\end{aligned}
\tag{1}
$$

### 3.1 NEURAL COLLAPSE

As a direct measurement of the terminal phase of training (TPT), neural collapse (Papyan et al., 2020) has been characterized by four properties in the last-layer features and class weight vectors:

- **Variability collapse.** The variance in last-layer features within the class progressively diminishes.
- **Convergence to simplex ETF.** Upon centering the global mean of the class means, the class means converge to a simplex ETF.
- **Convergence to self-duality.** Allowing for rescaling, the class means and their respective class weight vectors converge, thereby creating a complete and balanced symmetry determined by the classifier.
- **Simplification to nearest class center (NCC).** Based on the aforementioned properties, classifying the last-layer features by the nearest class mean produces the same outcome as the decision made by the classifier.

Among four manifestations that characterize the neural collapse, the main point is that the class means and class weight vectors collapse to each other at the convergence, and the shape of these points becomes a simplex ETF.

**Definition 1** (Simplex ETF). *A standard simplex ETF is a set of points in $\mathbb{R}^K$ composed by the columns of*

$$
\boldsymbol{M}^* = \sqrt{\frac{K}{K-1}} \left(\boldsymbol{I}_K - \frac{1}{K}\mathbf{1}_K\mathbf{1}_K^\intercal\right)
\tag{2}
$$

*where $\boldsymbol{I}_K \in \mathbb{R}^{K \times K}$ is the identity matrix and $\mathbf{1}_K \in \mathbb{R}^K$ is the ones vector.*

As the neural collapse, class weight vectors of $\mathbf{W}$ in Eq. 1 converge to the matrix which consists of the columns of $\alpha \mathbf{U} \boldsymbol{M}^* \in \mathbb{R}^{D \times K}$, where $\alpha \in \mathbb{R}_{\geq 0}$ is a scale factor and $\mathbf{U} \in \mathbb{R}^{D \times K}$ is a partial orthogonal matrix ($\mathbf{U}^\intercal \mathbf{U} = \boldsymbol{I}_K$).

### 3.2 LAYER-PEELED MODEL

Currently, most studies focus on the widely used cross entropy loss function in classification models. It is defined as:

$$
\mathcal{L}_{ce}\left(\mathbf{h}, \mathbf{W}\right) = -\log\left(\frac{\exp(\mathbf{h}^\intercal \mathbf{w}_c)}{\sum_{k=1}^K \exp(\mathbf{h}^\intercal \mathbf{w}_k)}\right).
\tag{3}
$$

However, analyzing neural networks presents difficulties due to its non-convexity and it invokes the need of a simplification for tractable analysis. In studying neural collapse, recent works typically concentrate exclusively on the last-layer features and class weight vectors as learnable components disregarding the encoder. It is termed as layer-peeled model (LPM) (Fang et al., 2021), and can be formulated as:

$$
\begin{aligned}
\min_{\mathbf{W}, \mathbf{H}} \quad & \frac{1}{N}\sum_{k=1}^K \sum_{i=1}^{n_k} \mathcal{L}_{ce}\left(\mathbf{h}_{k,i}, \mathbf{W}\right), \\
s.t. \quad & \|\mathbf{w}_k\|^2 \leq E_W, \forall 1 \leq k \leq K, \\
& \|\mathbf{h}_{k,i}\|^2 \leq E_H, \forall 1 \leq k \leq K, 1 \leq i \leq n_k,
\end{aligned}
\tag{4}
$$

Table 1: Comparison between Neural Collapse (NC) and Zero-mean Neural Collapse (ZNC). ($\langle\cdot,\cdot\rangle$: inner product, $A \to B$: $A$ converges to $B$, $\|\cdot\|$: $\ell_2$ norm, $\text{Avg}_i$: the average of all values according to the index $i$)

| | NC | ZNC |
|---|---|---|
| Class mean ($\boldsymbol{\mu}_k$) | $\frac{1}{N_k}\sum_{i=1}^{N_k} \boldsymbol{h}_{i,k}$ | |
| Global mean ($\boldsymbol{\mu}_G$) | $\frac{1}{K}\sum_{k=1}^{K}\boldsymbol{\mu}_k$ | $\mathbf{0}_D$ |
| Total covariance ($\Sigma_T$) | $\text{Avg}_{i,k}\left(\boldsymbol{h}_{i,k}-\boldsymbol{\mu}_G\right)\left(\boldsymbol{h}_{i,k}-\boldsymbol{\mu}_G\right)^{\mathsf{T}}$ | $\text{Avg}_{i,k}\left(\frac{1}{N_k}\sum_{i=1}^{N_k}\boldsymbol{h}_{i,k}\boldsymbol{h}_{i,k}^{\mathsf{T}}\right)$ |
| Between-class covariance ($\Sigma_B$) | $\text{Avg}_k\left(\boldsymbol{\mu}_k-\boldsymbol{\mu}_G\right)\left(\boldsymbol{\mu}_k-\boldsymbol{\mu}_G\right)^{\mathsf{T}}$ | $\text{Avg}_k\left(\boldsymbol{\mu}_k\boldsymbol{\mu}_k^{\mathsf{T}}\right)$ |
| Within-class covariance ($\Sigma_W$) | $\text{Avg}_{i,k}\left(\boldsymbol{h}_{i,k}-\boldsymbol{\mu}_k\right)\left(\boldsymbol{h}_{i,k}-\boldsymbol{\mu}_k\right)^{\mathsf{T}}$ | |
| *Variability collapse* | $\boldsymbol{\Sigma}_W \to \mathbf{0}_D$ | |
| *Convergence to a specific form* | $\big\|\|\boldsymbol{\mu}_k-\boldsymbol{\mu}_G\|-\|\boldsymbol{\mu}_{k'}-\boldsymbol{\mu}_G\|\big\| \to 0$ $\langle\tilde{\boldsymbol{\mu}_k},\tilde{\boldsymbol{\mu}_{k'}}\rangle \to \frac{K}{K-1}\delta_{k,k'}-\frac{K}{K-1}$ $\tilde{\boldsymbol{\mu}}_k = \left(\boldsymbol{\mu}_k-\boldsymbol{\mu}_G\right)/\|\boldsymbol{\mu}_k-\boldsymbol{\mu}_G\|$ | $\big\|\|\boldsymbol{\mu}_k\|-\|\boldsymbol{\mu}_{k'}\|\big\| \to 0$ $\langle\tilde{\boldsymbol{\mu}_k},\tilde{\boldsymbol{\mu}_{k'}}\rangle \to 0$ $\tilde{\boldsymbol{\mu}}_k = \boldsymbol{\mu}_k/\|\boldsymbol{\mu}_k\|$ |
| *Convergence to self-duality* | $\left\|\frac{\boldsymbol{W}^{\mathsf{T}}}{\|\boldsymbol{W}\|_F}-\frac{\dot{\boldsymbol{M}}}{\|\dot{\boldsymbol{M}}\|_F}\right\|_F \to 0$ $\dot{\boldsymbol{M}} = [\boldsymbol{\mu}_k-\boldsymbol{\mu}_G, 1\le k\le K]$ | $\left\|\frac{\boldsymbol{W}^{\mathsf{T}}}{\|\boldsymbol{W}\|_F}-\frac{\dot{\boldsymbol{M}}}{\|\dot{\boldsymbol{M}}\|_F}\right\|_F \to 0$ $\dot{\boldsymbol{M}} = [\boldsymbol{\mu}_k, 1\le k\le K]$ |
| *Simplification to NCC* | $\arg\max_{k'}\langle\boldsymbol{w}_{k'},\boldsymbol{h}\rangle+b_{k'}$ $\to \arg\min_{k'}\|\boldsymbol{h}-\boldsymbol{\mu}_{k'}\|$ | $\arg\max_{k'}\langle\boldsymbol{w}_{k'},\boldsymbol{h}\rangle$ $\to \arg\min_{k'}\|\boldsymbol{h}-\boldsymbol{\mu}_{k'}\|$ |

where $n_k$ is the number of samples in $k$-th class, and $E_H$ and $E_W$ are the $\ell_2$ norm constraints for feature $\mathbf{h}$ and class weight vector $\mathbf{w}$, respectively. $\|\cdot\|$ indicates $\ell_2$ norm.

Although the LPM model is not applicable in practice, it functions as an analytical tool and applies the learning behaviors of the last-layer features and class weight vectors in classification models. Actually, the encoder learning is through the multiplication between the Jacobian and the gradient with respect to the last-layer features, *i.e.*, $\frac{\partial f}{\partial \theta}\frac{\partial \mathbf{H}}{\partial f}\frac{\partial \mathcal{L}}{\partial \mathbf{H}}$. The LPM in Eq. 4 has been proven to achieve global optimality when satisfying neural collapse properties in balanced datasets (Graf et al., 2021) and imbalanced datasets (Yang et al., 2022b; Fang et al., 2021).

## 4 ZERO-MEAN NEURAL COLLAPSE

Although researchers have remarked on neural collapse and its theoretical analysis, the training dynamics of the fixed orthogonal classifier have not been deeply analyzed in terms of neural collapse. One possible reason is the geometric limitation of orthogonal classifiers which have fixed orthogonal matrices as their weight parameters so it cannot be a simplex ETF. To overcome the limitation and examine the appearance of collapse between last-layer features and class weight vectors in training the classification model with a fixed orthogonal classifier, we propose a *zero-mean neural collapse* that centers class means to the origin instead of the global mean.

- **Variability collapse, convergence to self-duality, and simplification to nearest class center (NCC).** The only difference is that class means are centered to the origin as still satisfying the properties of neural collapse.

- **Convergence to non-negative orthogonal matrix.** The class means progress to have orthogonality in the non-negative Euclidean space, as aligned to the fixed orthogonal classifier.

As considering orthogonality and centering class means to the origin, the main calculations for neural collapse should be changed. We summarized them in Table 1, and please refer to Appendix B for more details. Furthermore, we analyzed the zero-mean neural collapse in the same environments to neural collapse by conducting comprehensive experiments (Papyan et al., 2020) (Appendix B.1). As a result, we have more reliability in the fact that the zero-mean neural collapse is a natural phenomenon in training classification models with fixed non-negative orthogonal classifiers in TPT.

## 5  FIXED NON-NEGATIVE ORTHOGONAL CLASSIFIER

From the analysis of neural collapse using the LPM (Yang et al., 2022b), it is not guaranteed that the collapse phenomenon between class means and their respective class weight vectors occurs when the LPM with a fixed classifier (not a simplex) has the global optimality if no additional constraints are present. To invoke the collapse phenomenon between class means and their respective class weight vectors in fixed classifiers, it has been observed that imposing two constraints is highly beneficial for obtaining the global optimality and max-margin in decision with fixed classifiers: non-negativity and orthogonality. Concretely, we initialize the linear classifier $\mathbf{Q}$ as a random fixed non-negative orthogonal matrix by Eq. 8 and only optimize the last-layer features $\mathbf{H}$. In this case, the layer-peeled model with a fixed non-negative orthogonal classifier (OLPM) reduces to the following problem:

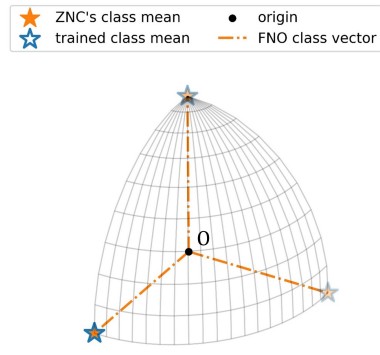

Figure 2: Visualization of Fixed Non-negative Orthogonal Classifier.

$$\min_{\mathbf{H}} \quad \frac{1}{N} \sum_{k=1}^{K} \sum_{i=1}^{n_k} \mathcal{L}_{ce}\left(\mathbf{h}_{k,i}, \mathbf{Q}^*\right),$$

$$s.t. \ \|\mathbf{h}_{k,i}\|^2 \leq E_H \text{ and } \sum_{j \neq k}^{K} \mathbf{h}_{k,i}^\mathsf{T} \mathbf{q}_j^* \geq 0, \ \forall 1 \leq k \leq K, \ 1 \leq i \leq n_k,$$

(5)

where $\mathbf{Q}^*$ is the fixed classifier as a non-negative orthogonal matrix and satisfies:

$$\mathbf{q}_k^{*\mathsf{T}} \mathbf{q}_{k'}^* = \delta_{k,k'}, \ \forall k, k' \in [1, K],$$

(6)

where all elements of the weight matrix is non-negative and $\delta_{k,k'}$ is the Kronecker delta function.

We have proved the global optimality for OLPM by maintaining a similar proof structure to Theorem 1 in Yang et al. (2022b) for easier understanding and comparison.

**Theorem 1.** *When any global minimizer* $\mathbf{H}^* = [\mathbf{h}_{k,i}^* : 1 \leq k \leq K, 1 \leq i \leq n_k]$ *of Eq. 5 is a max-margin solution, the global minimizer converges to a non-negative orthogonal matrix with the same direction as* $\mathbf{Q}^*$ *and a length of* $K-1$, *i.e.,*

$$\mathbf{h}_{k,i}^{*\mathsf{T}} \mathbf{q}_{k'}^* = (K-1)\delta_{k,k'}, \forall 1 \leq k, k' \leq K, 1 \leq i \leq n_k,$$

(7)

*which means that the zero-mean neural collapse phenomenon emerges.*

**Proof 1.** Please refer to Appendix C for our proof.

**Remark 1.** *As a fixed classifier cannot change the shape of weight matrix, it is impossible for the fixed classifier to satisfy the neural collapse if their shape is not a simplex ETF. However, we proposed the zero-mean neural collapse that considers non-negativity and orthogonality in the weight matrix and Theorem 1 shows that LPM with a fixed non-negative orthogonal classifier can inherently produce the zero-mean neural collapse solution even in inducing the max-margin in decision.*

Finally, we proposed a fixed non-negative orthogonal (FNO) classifier, which has an orthogonal matrix as their weight parameters. The FNO classifier satisfies all constraints of zero-mean neural collapse while locating on the non-negative Euclidean space.

**Definition 2** (Non-negative Orthogonal Classifier)**.** *A non-negative orthogonal classifier has a partial orthogonal weight matrix* $\mathbf{Q} \in \mathbb{R}_{\geq 0}^{D \times K}$, *which satisfies below properties:*

$$\mathbf{Q}^\mathsf{T}\mathbf{Q} = \boldsymbol{I}_K, \quad s.t. \ Q_{i,j} \geq 0, \ \forall 1 \leq i \leq D, 1 \leq j \leq K,$$

(8)

*where* $\boldsymbol{I}_n \in \mathbb{R}^{n \times n}$ *is the identity matrix and* $Q_{i,j}$ *is the* $(i, j)$ *element of* $\mathbf{Q}$.

Building on the theoretical benefits of the fixed non-negative orthogonal classifier in the OLPM model, additional advantages appear to empirical practices by using *feature dimension split* naturally induced by the non-negativity and orthogonality of our classifier, as illustrated in Figure 3.

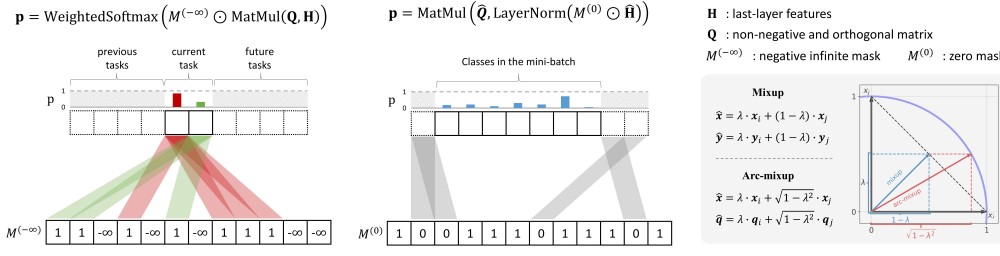

(a) FNO in Continual Learning        (b) FNO in Imbalanced Learning

Figure 3: Visualization of the application of the Fixed Non-negative Orthogonal (FNO) classifier in continual learning and imbalanced learning. (a) *Why continual learning?* FNO classifier is used with masked and weighted softmax to reduce the class-wise interference by enhancing the impact of masked softmax in continual learning. (b) *Why imbalanced Learning?* FNO classifier is utilized with arc-mixup and zero masking in mini-batches, resulting in superior performance in imbalanced learning.

## 6 THE BENEFITS OF FEATURE DIMENSION SEPARATION

*Feature separation* has been commonly used in various ways in the academia by dividing features into low-dimensional vectors while maintaining their values (Zuo et al., 2005; Miao et al., 2012; Kim et al., 2023b). The effect of *Feature Dimension Separation* (FDS) resulting from the FNO classifier is quite similar to feature separation and thus the naming as FDS is proper. To avoid confusion regarding this term, we specify FDS used in this paper as below.

**Definition 3** (Feature Dimension Separation). *Let $q_k = \{q_j\}_{1 \le j \le D}$ the $k$-th class weight vector in the fixed non-negative orthogonal classifier and $\mathbb{J}_k = \{j \mid q_j > 0, 1 \le j \le D\}$ an index set of $q_k$ where $q_j$ is not zero. Then, as the definition of FNO classifier, any index set of class weight vectors has disjoint to any other class weight vector and we call this phenomenon as feature dimension separation, i.e.,*

$$\mathbb{J}_k \cap \mathbb{J}_{k'} = \emptyset, \ \ \forall k \ne k', \tag{9}$$

*which means that the features' elements used for deciding the confidence to any class lose their utility to any other classes.*

Upon definition 3, input samples are classified by the classifier that only considers their own indices due to FDS, which is close to feature separation used in Kim et al. (2023b). However, FDS still entangles feature groups by using softmax and layer normalization, while the feature separation is unable to entangle them by separately using such normalization techniques.

### 6.1 ENHANCEMENT OF MASKED SOFTMAX IN CONTINUAL LEARNING

The softmax function invokes max-margin in the decision space and tends to induce the catastrophic forgetting in continual learning (Kim et al., 2023a; Mukhoti et al., 2022; Kendall & Gal, 2017). To resolve this negative influence of softmax, masked softmax (Kim et al., 2023a) has been proposed:

**Definition 4** (Masked Softmax (Kim et al., 2023a)). *To remove the class-wise interference of specific classes in softmax, the masked softmax multiplies a negative infinity mask $\boldsymbol{M}^{(-\infty)}$ to output vectors. When getting rid of $k$-th class's interference from $i$-th input sample, $k$-th element of the output vector is multiplied by negative infinite values, i.e.,*

$$\boldsymbol{M}_i^{(\cdot\infty)} = (m_j)_{1 \le j \le K}$$
$$\mathbf{p} = \text{SOFTMAX}\left(\boldsymbol{M}_i^{(\cdot\infty)} \odot (\mathbf{W}^\intercal \mathbf{h} + \mathbf{b})\right), \tag{10}$$

*where $m_j = -\infty$ if $j = k$ otherwise 1 and $\odot$ means the Hadamard product.*

Therefore, the class weight vectors corresponding to the masked classes remained unchanged during fine-tuning due to the negative infinity mask. However, the masked outputs still have the potential to change because all elements of them are entangled regardless of whether their class is masked

or not (Yang et al., 2023a). In contrast, the FNO classifier with masked softmax is less suffered such changeable situation as the class weight matrix becomes linearly independent due to FDS. Consequently, FDS enhances the masked softmax in reducing the class-wise interference.

Building on this enhancement, we exploited weighted softmax after masking, which utilizes an additional scale parameter, because the softmax is sensitive to the scale of input and fixed weight parameters cannot re-scale it as already known in fixed classifiers (Hoffer et al., 2018). Finally, we propose *masked and weighted softmax with the FNO classifier in the task $T_t$* and the total process of it is explained in Algorithm 1 (please refer to Appendix D).

## 6.2 ARC-MIXUP WITH FEATURE MASKING IN IMBALANCED LEARNING

Mixup (Zhang et al., 2017) is an effective augmentation method in imbalanced learning by linearly interpolating input samples (Zhong et al., 2021). However, the linear interpolation does not work well in the case that all class weight vectors and features are located on the hypersphere as shown in Figure 3b. To address this problem of mixup, we proposed a novel interpolation method *arc-mixup*.

**Definition 5** (Arc-mixup). *Arc-mixup is an interpolation-based method when all last-layer features and class weight vectors are located on the same hypersphere, while keeping the scale of mixed class weight vector $\hat{\mathbf{q}}$, i,e,.*

$$\hat{\boldsymbol{x}} = \lambda \cdot \boldsymbol{x}_i + \sqrt{1 - \lambda^2} \cdot \boldsymbol{x}_j$$
$$\hat{\mathbf{q}} = \lambda \cdot \mathbf{q}_{c_i} + \sqrt{1 - \lambda^2} \cdot \mathbf{q}_{c_j} \tag{11}$$
$$\mathcal{L}_{cls}(\hat{\boldsymbol{x}}, \hat{\mathbf{q}}) = -\log \hat{\mathbf{q}}^{\mathsf{T}} \hat{\mathbf{h}},$$

*where $\hat{\mathbf{h}}$ is the last-layer feature of $\hat{\boldsymbol{x}}$. This means that $\hat{\mathbf{q}}$ is still located on the hypersphere.*

As definition 5, it is not necessary to mixup input samples according to the rule of arc-mixup. However, we have empirically found that using the same mixup rate for both $\hat{\boldsymbol{x}}$ and $\hat{\mathbf{q}}$ performed better than applying different rates.

To emphasize the effect of FDS, we prove that arc-mixup without interference between different class features is only possible using the FNO classifier in the following theorem.

**Theorem 2** (Arc-mixup without class-wise interference). *There is no class-wise interference in arc-mixup when each element is determined by only one-side of samples. To satisfy this constraint while the mixed class weight vector $\hat{\mathbf{w}}$ has the same scale of the original class weight vector, the class weight matrix should be non-negative and orthogonal, i.e.,*

$$\hat{\mathbb{J}} = \mathbb{J}_i \cup \mathbb{J}_j, \;\; and \;\; |\hat{\mathbb{J}}| = |\mathbb{J}_i| + |\mathbb{J}_j| \tag{12}$$
$$\|\hat{\mathbf{w}}\| = \|\mathbf{w}_{c_i}\| = \|\mathbf{w}_{c_j}\| = 1,$$

*where $\hat{\mathbf{w}}$ is the mixed class weight vector from the class weight vectors for $\boldsymbol{x}_i$ and $\boldsymbol{x}_j$ and $\hat{\mathbb{J}}$ is the index set of $\hat{\mathbf{w}}$.*

*Proof.* (Orthogonality) The $\ell_2$-norm of $\hat{\mathbf{w}}$ is derived as below.

$$\|\hat{\mathbf{w}}\| = \sqrt{\sum_{d=1}^{D} \left( \lambda w_{c_i,d} + \sqrt{1-\lambda^2} w_{c_j,d} \right)^2} = \sqrt{1 + 2\lambda\sqrt{1-\lambda^2} \sum_{d=1}^{D} w_{c_i,d} w_{c_j,d}} \tag{13}$$

$\|\hat{\mathbf{w}}\| = 1$ if and only if $\sum_{d=1}^{D} w_{c_i,d} w_{c_j,d} = 0$, which means $\mathbf{w}_{c_i}$ and $\mathbf{w}_{c_j}$ are orthogonal.

(Non-negativity) Upon the orthogonality, non-negativity is a natural result as the definition of FDS and Eq. 12. □

Building on the arc-mixup, we shed light on the imbalance problem in a mini-batch. When the datasets are extremely imbalanced, the probability of not including input samples in tail classes is high. We considered that such circumstance might have a similar problem to continual learning, *i.e.*, momentarily catastrophic forgetting occurs about the tail classes that did not appear in the mini-batch. Inspired by this idea, we finally propose *arc-mixup with the FNO classifier and feature masking in the mini-batch* and the total process of it is explained in Algorithm 2 (please refer to Appendix D).

Table 2: Classification results for standard continual learning benchmarks. The experiments on S-Tiny-ImageNet utilized 5 trials with random seeds while other experiments utilized 10 trials with random seeds. Best in bold. ($\mathcal{B}$: Buffer size, RM: Rehearsal-based Method, Clf: Classifier, $M$: negative infinite masking.) (Please refer to the performance of commonly used methods in Table 7 (Appendix F).) (Final Average Accuracy ↑ (%): $mean_{std}$)

| $\mathcal{B}$ | Method | | | S-MNIST | | S-CIFAR-10 | | S-CIFAR-100 | | S-Tiny-ImageNet | |
|---|---|---|---|---|---|---|---|---|---|---|---|
| | RM | Clf | $M$ | Class-IL | Task-IL | Class-IL | Task-IL | Class-IL | Task-IL | Class-IL | Task-IL |
| 200 | ER | FC | ✓ | $82.98_{1.03}$ | $98.13_{0.16}$ | $61.75_{6.07}$ | $91.39_{2.13}$ | $28.51_{0.44}$ | $68.51_{0.87}$ | $15.47_{0.67}$ | $44.11_{0.50}$ |
| | ER | FNO | ✓ | $84.26_{1.16}$ | $98.45_{0.19}$ | $63.84_{1.47}$ | $92.03_{0.52}$ | $\mathbf{32.43}_{0.58}$ | $71.34_{1.17}$ | $17.31_{0.74}$ | $44.76_{0.90}$ |
| | | | | +1.28 | +0.32 | +2.09 | +0.64 | +3.92 | +2.83 | +1.84 | +0.65 |
| | DER++ | FC | ✓ | $84.45_{0.88}$ | $99.03_{0.09}$ | $66.35_{1.52}$ | $93.17_{0.54}$ | $28.57_{1.11}$ | $74.02_{0.76}$ | $13.21_{0.56}$ | $49.75_{0.99}$ |
| | DER++ | FNO | ✓ | $\mathbf{86.27}_{0.88}$ | $\mathbf{99.11}_{0.08}$ | $\mathbf{67.53}_{1.25}$ | $\mathbf{93.98}_{0.39}$ | $30.70_{1.16}$ | $\mathbf{74.11}_{0.96}$ | $\mathbf{18.44}_{0.94}$ | $\mathbf{53.06}_{0.67}$ |
| | | | | +1.82 | +0.08 | +1.18 | +0.81 | +2.13 | +0.09 | +5.23 | +3.31 |
| 500 | ER | FC | ✓ | $89.35_{0.59}$ | $\mathbf{99.20}_{0.16}$ | $70.64_{1.28}$ | $94.22_{0.41}$ | $35.68_{0.89}$ | $74.77_{0.71}$ | $20.43_{0.38}$ | $53.21_{0.84}$ |
| | ER | FNO | ✓ | $\mathbf{89.42}_{0.72}$ | $99.16_{0.17}$ | $71.43_{0.95}$ | $94.38_{0.43}$ | $39.80_{0.68}$ | $76.52_{0.86}$ | $22.41_{0.57}$ | $52.60_{0.58}$ |
| | | | | +0.07 | -0.04 | +0.79 | +0.16 | +4.12 | +1.75 | +1.98 | -0.61 |
| | DER++ | FC | ✓ | $83.10_{1.22}$ | $99.08_{0.09}$ | $71.85_{3.76}$ | $94.28_{1.49}$ | $37.80_{0.92}$ | $80.52_{0.60}$ | $17.71_{0.58}$ | $59.86_{1.08}$ |
| | DER++ | FNO | ✓ | $86.75_{0.75}$ | $99.00_{0.10}$ | $\mathbf{74.77}_{0.66}$ | $\mathbf{95.56}_{0.16}$ | $\mathbf{40.81}_{0.70}$ | $80.61_{0.46}$ | $\mathbf{22.45}_{0.36}$ | $\mathbf{59.87}_{1.91}$ |
| | | | | +3.65 | -0.08 | +2.92 | +1.28 | +3.01 | +0.09 | +4.74 | +0.01 |
| 5120 | ER | FC | ✓ | $93.51_{0.60}$ | $99.38_{0.12}$ | $82.63_{1.34}$ | $96.45_{0.27}$ | $52.95_{0.73}$ | $84.20_{0.58}$ | $35.73_{0.41}$ | $67.50_{0.53}$ |
| | ER | FNO | ✓ | $93.98_{0.39}$ | $99.47_{0.09}$ | $82.88_{1.35}$ | $96.79_{0.38}$ | $57.02_{0.52}$ | $85.66_{0.42}$ | $36.90_{0.41}$ | $66.86_{0.32}$ |
| | | | | +0.47 | +0.09 | +0.25 | +0.34 | +4.07 | +1.46 | +1.17 | -0.64 |
| | DER++ | FC | ✓ | $93.75_{0.23}$ | $\mathbf{99.62}_{0.05}$ | $84.71_{0.65}$ | $96.78_{0.16}$ | $58.18_{0.43}$ | $\mathbf{87.97}_{0.33}$ | $34.72_{0.46}$ | $72.40_{0.25}$ |
| | | DER++ | FNO | ✓ | $\mathbf{94.26}_{0.24}$ | $99.59_{0.05}$ | $\mathbf{85.65}_{0.38}$ | $\mathbf{97.20}_{0.13}$ | $\mathbf{58.82}_{0.43}$ | $87.35_{0.45}$ | $\mathbf{38.95}_{0.71}$ | $\mathbf{72.70}_{0.27}$ |
| | | | | +0.51 | -0.03 | +0.94 | +0.42 | +0.64 | -0.62 | +4.23 | +0.30 |

## 7 EXPERIMENTAL RESULTS

### 7.1 CONTINUAL LEARNING IN SPLIT DATASETS

**Implementation Details.** For equivalent experiments with Kim et al. (2023a), we followed Buzzega et al. (2020); Kim et al. (2023a) and conducted experiments on a split dataset of the MNIST by employing a full-connected network with two hidden layers, each one comprising of 100 ReLU units. For CIFAR-10, CIFAR100, and Tiny-ImageNet, we also use respective split datasets and rely on ResNet18 (He et al., 2015). For hyperparameter settings we exploited on our experiments, we follow Buzzega et al. (2020) where richly explored them for each datasets. We summarize the settings we used and please refer to Appendix E.1 for more explanations.

**Results and Analyses.** The comparison results of masked replay (Kim et al., 2023a) and our FNO classifier are illustrated on Table 2. The checkmark in $M$ means that they exploited original methods while negative infinity masking on last-layer features from current task but not on replayed samples. The FC and FNO indicate a learnable classifier and our FNO classifier, respectively. As shown in Table 2, the FNO classifier outperforms masked replay excepts few parts in task-incremental learning. This means that the FNO classifier enhances the masking effect by more effectively blocking the class-wise interference.

### 7.2 IMBALANCED LEARNING IN LONG-TAILED DATASETS

**Implementation Details.** We setup our experiments as following Zhong et al. (2021); Zhou et al. (2020) for CIFAR10-LT, ImageNet-LT, and Places-LT and following Yang et al. (2022b) for CIFAR100-LT. We exploited ResNet32 for CIFAR10-LT and increased the dimension of it 2 times than before for CIFAR100-LT. For ImageNet-LT and Places-LT, we used ResNet50 and ResNet152, respectively. For reproducing baselines to compare our classifier to other methods, we follow Zhong et al. (2021); Zhou et al. (2020) and set all hyperparameter settings are equivalent except the mixup alpha. Please refer to Appendix E.2 for more explanation and implementation details.

**Results and Analyses.** As shown in Tables 3 and 4, arc-mixup with our FNO classifier outperforms most of experiments and especially exhibits consistent improvement in highly imbalanced environ-

Table 3: Comparison Results for Imbalanced Learning on CIFAR10/100-LT. All experiments utilized 20 trials with random seeds. Best in bold. (Aug: Augmentation, Clf: Classifier, $\mathcal{L}$: Loss function, B-mixup: Balanced mixup, CE: Cross-Entropy loss) (†: equivalent experimental setup.) (Accuracy (%): $mean_{std}$)

| Method | | | Reference | CIFAR10-LT | | | CIFAR100-LT | | |
|---|---|---|---|---|---|---|---|---|---|
| Aug | Clf | $\mathcal{L}$ | | 100 | 50 | 10 | 100 | 50 | 10 |
| mixup | FC | CE | (Yang et al., 2022b) | $73.90_{0.30}$ | $79.30_{0.20}$ | $87.80_{0.10}$ | $43.00$ | $48.10$ | - |
| B-mixup | FC | CE | (Zhang et al., 2022b) | $78.70$ | - | $\mathbf{89.60}$ | - | - | - |
| mixup | ETF | CE | (Yang et al., 2022b) | $67.00_{0.40}$ | $77.20_{0.30}$ | $87.00_{0.20}$ | - | - | - |
| mixup | ETF | DR | (Yang et al., 2022b) | $76.50_{0.30}$ | $81.00_{0.20}$ | $87.70_{0.20}$ | $45.30$ | $50.40$ | - |
| mixup | FC | CE | (reproduced.)† | $74.24_{0.44}$ | $80.00_{0.54}$ | $89.08_{0.32}$ | $43.80_{0.42}$ | $49.57_{0.37}$ | $\mathbf{63.90}_{0.33}$ |
| mixup | ETF | DR | (reproduced.)† | $75.18_{0.49}$ | $80.17_{0.31}$ | $87.29_{0.23}$ | $45.45_{0.38}$ | $50.67_{0.37}$ | $62.84_{0.38}$ |
| arc-mixup | FNO | CE | (reproduced.)† | $\mathbf{82.59}_{0.26}$ | $\mathbf{85.13}_{0.25}$ | $89.50_{0.14}$ | $\mathbf{49.26}_{2.82}$ | $\mathbf{54.44}_{2.32}$ | $63.14_{3.82}$ |

Table 4: Results of Imbalanced Learning on ImageNet-LT and Places-LT. All experiments utilized 3 trials with random seeds. Best in bold. (Aug: Augmentation, Clf: Classifier, $\mathcal{L}$: Loss function, FT: using the backbone pretrained on ImageNet.) (†: equivalent experimental setup, *: different hyperparameter setting.) (Accuracy (%): $mean_{std}$)

| Method | | | Reference | ImageNet-LT (ResNet50) | | | | Places-LT | |
|---|---|---|---|---|---|---|---|---|---|
| Aug | Clf | $\mathcal{L}$ | | Many | Median | Few | All | ResNet152 | ResNet152 (FT) |
| mixup | FC | CE | (Yang et al., 2022b)* | - | - | - | $44.30$ | - | - |
| mixup | ETF | DR | (Yang et al., 2022b)* | - | - | - | $44.70$ | - | - |
| - | FC | CE | (reproduced)† | $66.53_{0.18}$ | $40.34_{0.45}$ | $12.03_{0.30}$ | $45.88_{0.31}$ | $22.62_{0.29}$ | $24.16_{0.41}$ |
| mixup | FC | CE | (reproduced)† | $\mathbf{67.40}_{0.55}$ | $38.74_{0.83}$ | $9.12_{0.40}$ | $45.03_{0.64}$ | $22.10_{0.27}$ | $24.83_{1.19}$ |
| mixup | ETF | DR | (reproduced.)† | $64.17_{0.27}$ | $22.12_{0.38}$ | $0.57_{0.16}$ | $34.96_{0.27}$ | $23.11_{0.16}$ | $25.51_{0.08}$ |
| arc-mixup | FNO | CE | (reproduced)† | $59.46_{0.51}$ | $\mathbf{43.54}_{0.26}$ | $\mathbf{24.48}_{0.55}$ | $\mathbf{46.60}_{0.40}$ | $\mathbf{30.07}_{0.40}$ | $\mathbf{34.06}_{0.10}$ |

ments. Furthermore, when comparing the performance of mixup on hypersphere with cross-entropy loss that inspired us, with and without mixup, we figure out that linear interpolation performs poorly and requires a special form of loss function. In contrast, our method performs well, even though it does not require such loss function and uses a simpler approach. Furthermore, mixup has improved their performance in long-tailed datasets by increasing the accuracy of input samples in head classes, which means that the mixup worsens the imbalanced effect in decision rather than solving it, while our method shows the opposite tendency as illustrated in Table 4. Please refer to Tables 9, 10 and 11 in Appendix F for more detail experimental results.

## 8 CONCLUSION AND FUTURE WORK

**Summary.** We offered the *zero-mean collapse* to explain the collapse between class means and class weight vectors by centering the class means to the origin and considering orthogonality in the non-negative Euclidean space. Furthermore, we proposed the *FNO classifier* satisfying the properties of zero-mean collapse. To demonstrate that zero-mean collapse emerges in training a classification model with the FNO classifier, we proved it by using an orthogonal layer-peeled model. We found that the class weight matrix in the FNO classifier becomes linearly independent, called *FDS* in this paper, due to their non-negativity and orthogonality, and utilized it to masked softmax and mixup for resolving problems in continual learning and imbalanced learning, respectively. As a result, we exhibited the enhancement of masking on output vectors by alleviating the class-wise interference due to FDS in continual learning. In imbalanced learning, we proposed the *arc-mixup*, a variant of mixup effective on the hypersphere, and exhibited its effectiveness with the FNO classifier.

**Future Work.** Although the FNO classifier presents several advantages, it does have a clear limitation: $D$ should be greater than or equal to $K$. However, Lu & Steinerberger (2020) has demonstrated a proposition that explains the collapse phenomenon when $K \to \infty$ with fixed $D$, *i.e.*, $D \ll K$. This proposition, Thomson problem, and Jiang et al. (2023) will be the key to addressing the limitations of not only ours but also other fixed simplex ETF methods.

## REPRODUCIBILITY STATEMENT

We summarize the reproducibility statement of this paper as below:

- **Section 4.** To reproduce formulations and analyses for the zero-mean neural collapse, we provide detailed experimental settings and results in Appendix B. The related codes are also offered in the supplementary material.
- **Section 5.** To prove Theorem 1, we demonstrate the detailed proof of the theorem in Appendix C.
- **Section 6.** To reproduce the methods derived from FDS, we provide algorithms for more detailed processes in Appendix D. The codes related to Algorithms 1 and 2 are also included in the supplementary material.
- **Section 7.** All experiments can be reproduced by our supplementary material, which have dataset explanation, model architectures, and hyperparameter settings in Appendix E and Appendix F, and our codes which include configuration files for each experiment. Experimental settings, such as requirement libraries, are also provided through the README files in the codes.

Furthermore, our codes are publicly available at *link*.

## ACKNOWLEDGMENTS

This work was partly supported by Institute of Information & communications Technology Planning & Evaluation (IITP) grant funded by the Korea government (MSIT) (No.2019-0-01842, Artificial Intelligence Graduate School Program (GIST)), the National Research Foundation of Korea (NRF) grant funded by the Korea government (MSIT) (No.2022R1A2C2012054, Development of AI for Canonicalized Expression of Trained Hypotheses by Resolving Ambiguity in Various Relation Levels of Representation Learning), and Culture, Sports and Tourism R&D Program through the Korea Creative Content Agency grant funded by the Ministry of Culture, Sports and Tourism in 2022 (Project Name: Development of service robot and contents supporting children's reading activities based on artificial intelligence, Project Number:R2022060001, Contribution Rate: 33.3%)

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

# A    RELATED WORK

We provided an elaborate treatment of related work.

## A.1    ANALYSES AND EXTENSIONS OF NEURAL COLLAPSE

Extensive research has examined the neural collapse from various perspectives, including loss function (Lu & Steinerberger, 2020; Wojtowytsch et al., 2020), the use of mse loss (Han et al., 2021), feature normalization (Yaras et al., 2022), label smoothing (Zhou et al., 2022b), fine-grained structure (Yang et al., 2023b), generalization (Hui et al., 2022), out-of-distribution with normalization (Haas et al., 2022), imbalanced learning (Dang et al., 2023; Zhong et al., 2023), transfer learning (Galanti et al., 2021; Kornblith et al., 2021), federated learning (Li et al., 2023) or complex of them (Xu et al., 2023). Additionally, some studies have considered extensions of neural collapse such as intermediate layer collapse (Rangamani et al., 2023), generalized neural collapse (Liu et al., 2023), a novel metric (Xu & Liu, 2023). Naturally, geometric analyses have also arisen (Zhu et al., 2021; Thrampoulidis et al., 2022; Tirer et al., 2023).

## A.2    ADVANTAGES OF ORTHOGONALITY

Orthogonality has also shown advantages in various environments, such as regularization in multi-task learning (He et al., 2020), vision applications (Cao et al., 2021), features normalization (Lu et al., 2023), optimization (Chiley et al., 2019; Hu et al., 2020; Wu et al., 2020), regularization, training stability, robustness, generalization (Liu et al., 2021; Achour et al., 2022; Xu et al., 2022; Wang et al., 2020; Li et al., 2019a; Ranasinghe et al., 2021; Trockman & Kolter, 2021), explainability (Zhang et al., 2022a; Yang et al., 2020b), continual learning (Pernici et al., 2021b; Hersche et al., 2022; Chaudhry et al., 2020; Ramasesh et al., 2021; Saha et al., 2021; Farajtabar et al., 2020), reduction on computation load (Yang et al., 2020a), quantization (Ma et al., 2023), graph neural networks (Bodnar et al., 2022; Yang et al., 2022a), transformer (Huang et al., 2022; Kong et al., 2022), representation learning (Medini et al., 2020; Tiao et al., 2023), disentanglement (Sarhan et al., 2020; Liu et al., 2020; Cha & Thiyagalingam, 2023).

## A.3    ADDITIONAL INTRODUCTION AND RELATED WORK.

**Fixed Classifier.**    The fully-connected (FC) layer followed by softmax and cross-entropy loss is a commonly used classification layer in neural networks (Dunne & Campbell, 1997; Li et al., 2019b; Kobayashi, 2019). However, prior works have revealed that exploiting a predefined (Park & Sandberg, 1991) or random (Huang et al., 2006) projection matrix instead of FC layer can allow the models to learn useful representations without modifying the classifier. As a result, the need for the learnable classifier has led to the development of the fixed classifier, which effectively reduces computational costs and demonstrates competitive performance (Hoffer et al., 2018; Qian et al., 2020; Goyal et al., 2019). In addition to using special form matrices, such as orthogonal and Hadamard (Qian et al., 2020; Hoffer et al., 2018), fixed classifiers also incorporate regular polytopes (Coxeter, 1973), *e.g.*, a cube, orthoplex, and simplex (Pernici et al., 2019; 2021a; Mettes et al., 2019). However, the most of them demonstrated inferior performance except for simplex and orthoplex, which incorporate neural collapse and orthogonality, respectively (Pernici et al., 2019).

**Neural collapse and Orthogonality.**    The equiangular tight frame (ETF) is an optimal way of packing lines in Euclidean space. A regular simplex is a special type of ETF where there are one more vectors than the dimension of the space they span (Fickus et al., 2021; 2018). In neural collapse, class means of last-layer features and classifiers gradually approach each other and eventually form a simplex ETF in training classification model with learnable classifiers in TPT (Papyan et al., 2020). Extensive research has examined neural collapse from various perspectives and some studies have considered extensions of neural collapse. Naturally, geometric analyses have also arisen (Appendix A.1). Among them, many works have demonstrated that neural collapse emerges when the layer-peeled model (Fang et al., 2021), a useful simplification model that considers only last-layer features and class weight vectors by peeling away previous layers, achieves global optimality with various or no constraints (Nguyen et al., 2022; Kothapalli et al., 2022; Ji et al., 2022; Mixon et al., 2020; Tirer & Bruna, 2022). Additionally, fixed simplex ETF classifiers

have been proposed (Liang & Davis, 2023) and have performed better in continual learning (Yang et al., 2023a) and imbalanced learning (Thrampoulidis et al., 2022; Yang et al., 2022b). In contrast, although orthogonality has also shown advantages in various environments (Appendix A.2), it has not been deeply studied about the collapse phenomenon in training classification models with the fixed orthogonal classifier. Likewise, the fixed orthogonal classifier has not been utilized in continual learning and imbalanced learning although they have demonstrated the effectiveness in a similar way to the simplex ETF (Hoffer et al., 2018; Pernici et al., 2019; 2021a;b). To the best of our knowledge, the reason for this is that enforcing orthogonality results in the classifier being orthogonal to each other while class means centering to their global mean converge to a simplex ETF. A few research have analyzed neural collapse with orthogonality by using LPM with positive features (Nguyen et al., 2022) and exploiting unconstrained features model (Tirer & Bruna, 2022), but they did not freeze the classifier and did not consider non-negativity and orthogonality in the classifier simultaneously. RBL with an orthogonal-constrained classifier (Peifeng et al., 2023) has also shown the effectiveness of orthogonality in imbalanced learning. However, their classifier was not analyzed with LPM and did not take into account non-negativity, which is an important factor that causes feature dimension separation in the weight matrix of the fixed orthogonal classifier. The findings of neural collapse under MSE loss (Zhou et al., 2022a) yield similar results to ours. Specifically, the class means and class weight vectors collapse in orthogonal shape. However, their classifier was not fixed and the cross entropy loss, which is the most widely used loss function in classification models, was not utilized. These differences highlight the improved utility of our work.

Based on the different geometric feature of orthogonality, we have developed methods with the conviction that a fixed orthogonal classifier has potential in both continual learning and imbalanced learning much like a fixed simplex ETF, despite it not converging to a simplex ETF. With the intuition, we examined the collapse phenomenon that occurs when training a classification model with the fixed orthogonal classifier in the non-negative Euclidean space during TPT, which we have termed *zero-mean neural collapse*. We then proposed a *fixed non-negative orthogonal classifier* that satisfies the properties of zero-mean neural collapse while also providing benefits in continual learning and imbalanced learning by class-wise separation of the last-layer feature dimensions.

## B   ZERO-MEAN NEURAL COLLAPSE

**Class Mean.**   In the common neural collapse, the trained class mean is as Eq. 14.

$$\boldsymbol{\mu}_k = \frac{1}{N_k} \sum_{i=1}^{N_k} \boldsymbol{h}_{i,k} \tag{14}$$

where $K$ is the number of classes and $N_k$ is the number of input samples included in the $k$-th class. $\boldsymbol{h}_{i,k} \in \mathbb{R}^D$ means the feature vector of $i$-th input sample in the $k$-th class.

The class mean in ZNC $\boldsymbol{\mu}_k^0$ is calculated in the same way with the trained class mean Eq. 15.

$$\boldsymbol{\mu}_k^0 = \boldsymbol{\mu}_k \tag{15}$$

**Global Mean.**   In the common neural collapse, the trained global mean $\boldsymbol{\mu}_G$ is as Eq. 16.

$$\boldsymbol{\mu}_G = \frac{1}{K} \sum_{k=1}^{K} \boldsymbol{\mu}_k \tag{16}$$

where $\boldsymbol{\mu}_k$ is the $k$-th class mean.

This global mean will not be trained and fixed to zero-mean vector like Eq. 17.

$$\boldsymbol{\mu}_G^0 = \mathbf{0}_D \tag{17}$$

where $\boldsymbol{\mu}_G^0$ denotes the global mean of ZNC and $\mathbf{0}_D \in \mathbb{R}^D$ is the zeros vector.

**Total Covariance.**   In the common neural collapse, the trained total covariance $\boldsymbol{\Sigma}_T \in \mathbb{R}^{D \times D}$ is as Eq. 18.

$$\boldsymbol{\Sigma}_T = \frac{1}{K} \sum_{k=1}^{K} \left( \frac{1}{N_k} \sum_{i=1}^{N_k} \left( \boldsymbol{h}_{i,k} - \boldsymbol{\mu}_G \right) \left( \boldsymbol{h}_{i,k} - \boldsymbol{\mu}_G \right)^\mathsf{T} \right) \tag{18}$$

This total covariance will be changed like Eq. 19.

$$\boldsymbol{\Sigma}_T^0 = \frac{1}{K}\sum_{k=1}^{K}\left(\frac{1}{N_k}\sum_{i=1}^{N_k}\boldsymbol{h}_{i,k}\boldsymbol{h}_{i,k}^\intercal\right)\quad\left(\because\boldsymbol{\mu}_G^0 = \boldsymbol{0}_D\right) \tag{19}$$

where $\boldsymbol{\Sigma}_T^0\in\mathbb{R}^{D\times D}$ denotes the total covariance in the ZNC.

**Between-class Covariance.** In the common neural collapse, the between-class covariance $\boldsymbol{\Sigma}_B\in\mathbb{R}^{D\times D}$ is as Eq. 20.

$$\boldsymbol{\Sigma}_B = \frac{1}{K}\sum_{k=1}^{K}\left((\boldsymbol{\mu}_k - \boldsymbol{\mu}_G)(\boldsymbol{\mu}_k - \boldsymbol{\mu}_G)^\intercal\right) \tag{20}$$

This between-class covariance will be changed like Eq. 21.

$$\boldsymbol{\Sigma}_B^0 = \frac{1}{K}\sum_{k=1}^{K}\boldsymbol{\mu}_k\boldsymbol{\mu}_k^\intercal\quad\left(\because\boldsymbol{\mu}_G^0 = \boldsymbol{0}_D\right) \tag{21}$$

where $\boldsymbol{\Sigma}_B^0\in\mathbb{R}^{D\times D}$ denotes the between-class covariance in the ZNC.

**Within-class Covariance.** In the common neural collapse, the within-class covariance $\boldsymbol{\Sigma}_W\in\mathbb{R}^{D\times D}$ is as Eq. 22.

$$\boldsymbol{\Sigma}_W = \frac{1}{K}\sum_{k=1}^{K}\left(\frac{1}{N_k}\sum_{i=1}^{N_k}(\boldsymbol{h}_{i,k} - \boldsymbol{\mu}_k)(\boldsymbol{h}_{i,k} - \boldsymbol{\mu}_k)^\intercal\right) \tag{22}$$

The zero within-class covariance $\boldsymbol{\Sigma}_W^0$ is calculated in the same way with the within-class covariance (Eq. 23.

$$\boldsymbol{\Sigma}_W^0 = \boldsymbol{\Sigma}_W \tag{23}$$

where $\boldsymbol{\Sigma}_W^0\in\mathbb{R}^{D\times D}$ denotes the within-class covariance in the ZNC.

**(NC1) Variability collapse:** $\boldsymbol{\Sigma}_W \to \boldsymbol{0}_D$

**(ZNC1) Variability collapse:** $\boldsymbol{\Sigma}_W^0 \to \boldsymbol{0}_D$

**(NC2) Convergence to simplex ETF:**

$$\left|\|\boldsymbol{\mu}_k - \boldsymbol{\mu}_G\| - \|\boldsymbol{\mu}_{k'} - \boldsymbol{\mu}_G\|\right| \to 0 \quad \forall_{k,k'}$$

$$\langle\tilde{\boldsymbol{\mu}}_k, \tilde{\boldsymbol{\mu}}_{k'}\rangle \to \frac{K}{K-1}\delta_{k,k'} - \frac{K}{K-1} \quad \forall_{k,k'}$$

where $\tilde{\boldsymbol{\mu}}_k = (\boldsymbol{\mu}_k - \boldsymbol{\mu}_G)/\|\boldsymbol{\mu}_k - \boldsymbol{\mu}_G\|$ denotes the normalized $k$-th class mean and $\delta_{k,k'}$ is the Kronecker delta function.

**(ZNC2) Convergence to orthogonal matrix:**

$$\left|\|\boldsymbol{\mu}_k\| - \|\boldsymbol{\mu}_{k'}\|\right| \to 0 \quad \forall_{k,k'}$$

$$\langle\tilde{\boldsymbol{\mu}}_k, \tilde{\boldsymbol{\mu}}_{k'}\rangle \to 0 \quad \forall_{k,k'}$$

where $\tilde{\boldsymbol{\mu}}_k = \boldsymbol{\mu}_k/\|\boldsymbol{\mu}_k\|$ denotes the normalized $k$-th class mean.

**(NC3) Convergence to self-duality:**

$$\left\|\frac{\boldsymbol{W}^\intercal}{\|\boldsymbol{W}\|_F} - \frac{\dot{\boldsymbol{M}}}{\|\dot{\boldsymbol{M}}\|_F}\right\|_F \to 0$$

where $\dot{\boldsymbol{M}} = [\boldsymbol{\mu}_k - \boldsymbol{\mu}_G, 1\le k\le K]\in\mathbb{R}^{D\times K}$ denotes the matrix obtained by concatenating the class means into the columns of a matrix.

Table 5: Test Accuracy (%) at zero error and last epoch.

| Arch \ Dataset | MNIST | | Fashion MNIST | | SVHN | | CIFAR10 | | CIFAR100 | |
|---|---|---|---|---|---|---|---|---|---|---|
| | Zero | Last | Zero | Last | Zero | Last | Zero | Last | Zero | Last |
| VGG | 99.40 | 99.56 | 92.92 | 93.31 | 93.82 | 94.53 | 87.85 | 88.65 | 63.03 | 63.85 |
| ResNet | 99.32 | 99.71 | 93.29 | 93.64 | 94.64 | 95.70 | 88.72 | 89.44 | 66.19 | 66.21 |
| DenseNet | 99.65 | 99.70 | 94.18 | 94.35 | 95.87 | 95.93 | 91.14 | 91.19 | 77.19 | 76.56 |
| VGG$^\dagger$ | 99.49 | 99.56 | 93.25 | 93.56 | 93.91 | 94.51 | 87.54 | 88.44 | 63.14 | 63.95 |
| ResNet$^\dagger$ | 99.52 | 99.64 | 93.96 | 93.77 | 92.45 | 95.21 | 87.49 | 89.45 | 66.03 | 65.11 |
| DenseNet$^\dagger$ | 99.64 | 99.61 | 93.92 | 94.21 | 95.21 | 95.41 | 88.20 | 88.43 | 70.66 | 70.95 |
| VGG+FNO$^\dagger$ | 99.49 | 99.49 | 92.98 | 93.31 | 93.89 | 94.16 | 88.14 | 89.10 | 62.59 | 64.17 |
| ResNet+FNO$^\dagger$ | 99.40 | 99.67 | 93.90 | 94.28 | 92.30 | 95.41 | 87.61 | 89.59 | 67.15 | 66.20 |
| DenseNet+FNO$^\dagger$ | 99.62 | 99.60 | 93.91 | 94.08 | 95.11 | 95.22 | 88.64 | 88.76 | 72.23 | 72.54 |

**(ZNC3) Convergence to self-duality:**

$$\left\| \frac{\boldsymbol{W}^\intercal}{\|\boldsymbol{W}\|_F} - \frac{\dot{\boldsymbol{M}}}{\|\dot{\boldsymbol{M}}\|_F} \right\|_F \to 0$$

where $\dot{\boldsymbol{M}} = [\boldsymbol{\mu}_k, 1 \le k \le K] \in \mathbb{R}^{D \times K}$.

**(NC4) Simplification to NCC:**

$$\arg\max_{k'} \langle \boldsymbol{w}_{k'}, \boldsymbol{h} \rangle + b_{k'} \to \arg\min_{k'} \|\boldsymbol{h} - \boldsymbol{\mu}_{k'}\|$$

**(ZNC4) Simplification to NCC:**

$$\arg\max_{k'} \langle \boldsymbol{w}_{k'}, \boldsymbol{h} \rangle \to \arg\min_{k'} \|\boldsymbol{h} - \boldsymbol{\mu}_{k'}\|$$

## B.1 ZERO-MEAN NEURAL COLLAPSE IN IMAGE CLASSIFICATION BENCHMARKS

**Datasets.** To reproduce the analyses environment for zero-mean neural collapse, we follow Papyan et al. (2020). We utilize the MNIST (Deng, 2012), Fashion MNIST (Xiao et al., 2017), SVHN (Netzer et al., 2011), CIFAR10, and CIFAR100 datasets (Krizhevsky et al., 2009). To eliminate imbalance factor in the datasets, we subsamples 5,000 samples per class for MNIST and 4,600 samples per class for SVHN. The remaining datasets are already balanced: 6,000 examples for Fashion MNIST, 5,000 examples for CIFAR10, and 500 examples for CIFAR100. There was no data augmentation except normalization.

**Architectures and Implementation Details.** We train three types of convolutional networks: VGG (Simonyan & Zisserman, 2014), ResNet (He et al., 2015), and DenseNet (Huang et al., 2017). Following Papyan et al. (2020), we minimize the cross-entropy loss using SGD with momentum 0.9 and the weight decay is set to $5e^{-4}$. The batch size and the number of epochs are set to 256 and 350, respectively. The initial learning is set differently for datasets and architectures and it is decayed by 10 at 1/3 and 2/3. We itemize architectures for each dataset with the initial learning rate:

- **MNIST**: VGG11 (0.06786), ResNet18 (0.013296), and DenseNet40 (0.094015)
- **Fashion MNIST**: VGG11 (0.009597), ResNet18 (0.13025), and DenseNet250 (0.009597)
- **SVHN**: VGG11 (0.094015), ResNet18 (0.009597), and DenseNet40 (0.06786)
- **CIFAR10**: VGG13 (0.048982), ResNet18 (0.06786), and DenseNet40 (0.094015)
- **CIFAR100**: VGG13 (0.180451), ResNet50 (0.13025), and DenseNet250 (0.13025)

**Results and Analyses.** We visualize the observations of various experiments. Best descriptions in captions of each figure.

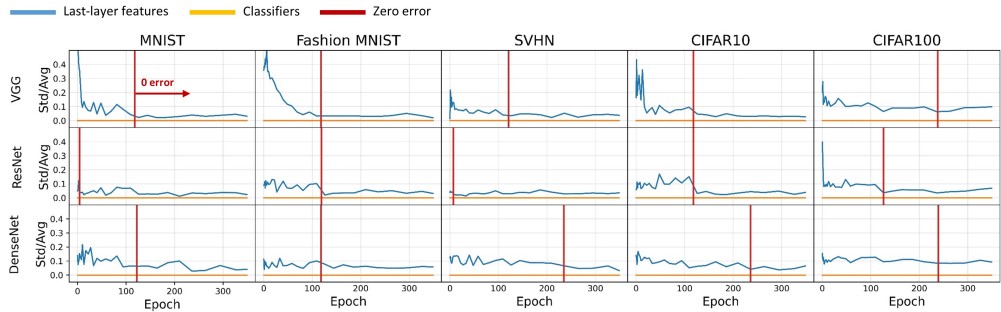

Figure 4: Class mean vectors become equinorm. The y-axis in each cell displays the coefficient of variation of averaged normalization of class means and class weight vectors. The blue lines represent $\mathrm{Std}_k \left( \|\boldsymbol{\mu}_k\| \right) / \mathrm{Avg}_k \left( \|\boldsymbol{\mu}_k\| \right)$ where $\boldsymbol{\mu}_k$, $1 \leq k \leq K$ denotes the class means of the last-layer features of the training input samples. The orange lines show $\mathrm{Std}_k \left( \|\boldsymbol{w}_k\| \right) / \mathrm{Avg}_k \left( \|\boldsymbol{w}_k\| \right)$ where $\boldsymbol{w}_k$ is the class weight vector of the $k$-th class. As training advances, the coefficients of variation for both class means and class weight vectors decrease.

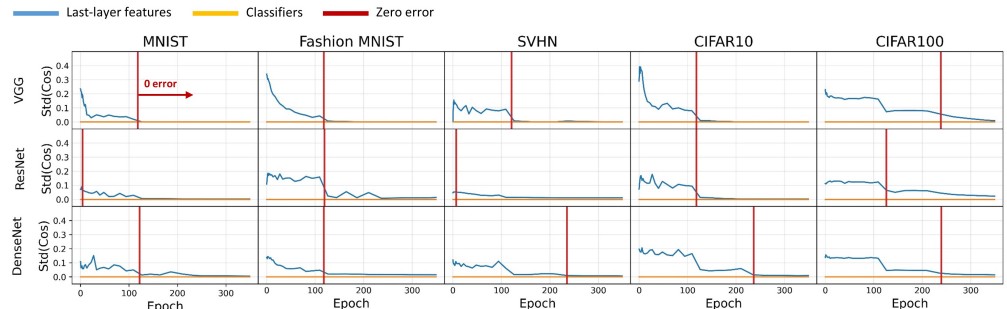

Figure 5: Class means converge to equiangularity. The y-axis in each cell displays the standard deviation of the cosine similarity between pairs of centered class means and class weight vectors across all pairs of classes $k$ and $k'$, $\forall k \neq k'$. $\boldsymbol{w}_k$, $\boldsymbol{\mu}_k$, and $\boldsymbol{\mu}_G$ are as in Figure 4. The blue lines represent $\cos_{\boldsymbol{\mu}} \left( k, k' \right) = \left( \boldsymbol{\mu}_k - \boldsymbol{\mu}_G \right) \left( \boldsymbol{\mu}_{k'} - \boldsymbol{\mu}_G \right)^{\mathsf{T}} / \left( \|\boldsymbol{\mu}_k - \boldsymbol{\mu}_G\| \|\boldsymbol{\mu}_{k'} - \boldsymbol{\mu}_G\| \right) \forall k \neq k'$. The orange lines show $\cos_{\boldsymbol{w}} \left( k, k' \right) = \boldsymbol{w}_k \boldsymbol{w}_{k'}^{\mathsf{T}} / \left( \|\boldsymbol{w}_k\| \|\boldsymbol{w}_{k'}\| \right) \forall k \neq k'$. As training advances, the standard deviations of $\cos_{\boldsymbol{\mu}} \left( k, k' \right)$ and $\cos_{\boldsymbol{w}} \left( k, k' \right)$ converge to zero, signifying equiangularity.

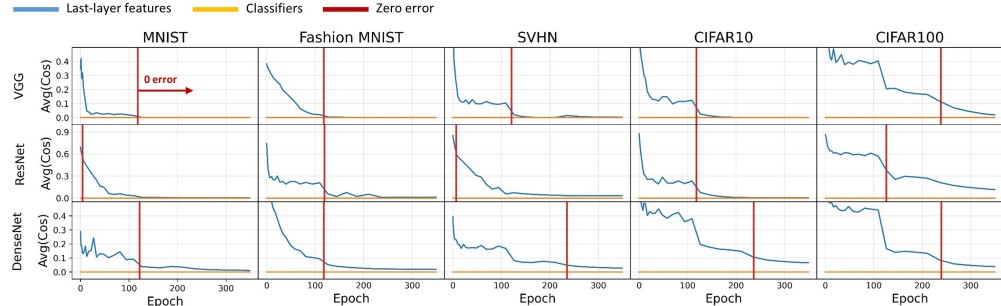

Figure 6: Class means converge to maximal-angle equiangularity. The y-axis in each cell displays the average of cosine similarity of between pairs of centered class means and between pairs of class weight vectors across all pairs of classes $k$ and $k'$, $\forall k \neq k'$. The blue lines represent $\mathrm{Avg}_{k,k'} \cos_{\boldsymbol{\mu}} \left( k, k' \right)$, $\forall k \neq k'$. The orange lines show $\mathrm{Avg}_{k,k'} \cos_{\boldsymbol{w}} \left( k, k' \right)$, $\forall k \neq k'$. This result represents the maximum separation achievable among globally centered, equiangular vectors.

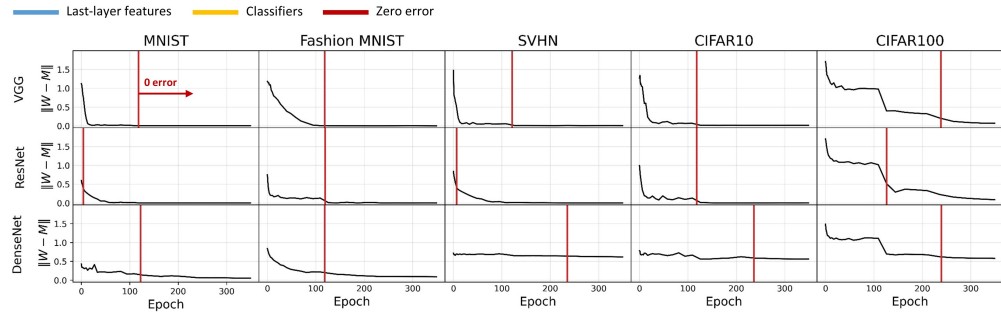

Figure 7: Class means approach to class weight vectors. The y-axis in each cell displays the distance between the class weight vectors and the normalized, centered class means. The lines represent the quantities of $\|\boldsymbol{W} - \tilde{\boldsymbol{M}}\|_F^2$. $\boldsymbol{W}$ is the class weight matrix. $\tilde{\boldsymbol{M}} = \dot{\boldsymbol{M}}/\|\dot{\boldsymbol{M}}\|_F$ where $\dot{\boldsymbol{M}} = (\boldsymbol{\mu}_k)_{1 \leq k \leq K} \in \mathbb{R}^{D \times K}$ is the matrix whose columns consist of the centered class means. This distance decreases as training advances. This result indicates that the centered class means are proportionally related to the class weight vectors like a self-dual manner.

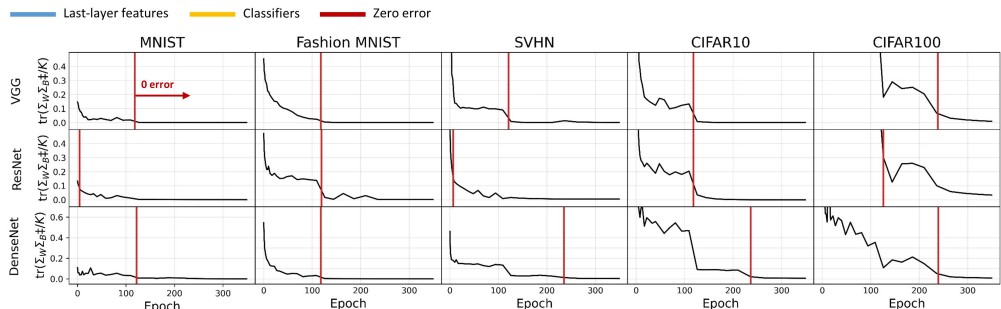

Figure 8: Within-class variation collapses. The y-axis in each cell displays the magnitude of the between-class covariance compared with the within-class covariance of the last-layer features. The lines represent $\mathrm{tr}(\Sigma_W \Sigma_B^{\ddagger}/K)$, where $\mathrm{tr}(\cdot)$ denotes the trace operator, $[\cdot]^{\ddagger}$ indicates Moore-Penrose pseudoinverse, and $K$ is the number of classes. $\Sigma_W$ and $\Sigma_B$ are as in Table 1. This magnitude decreases as training advances. This result indicates that collapse of within-class variations occurs.

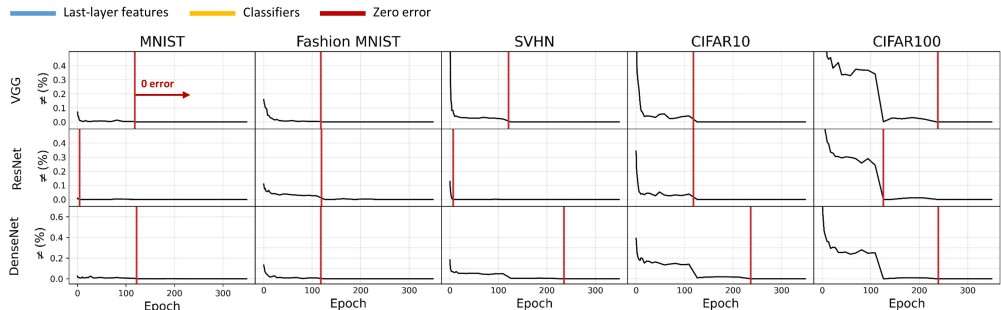

Figure 9: Classifier works in a similar way to NCC. The y-axis in each cell displays the percentage of input samples in the training set where there is a mismatch between the classifier;s output and the result that would have been obtained by selecting $\arg\min_k \|\boldsymbol{h} - \boldsymbol{\mu}_k\|$ where $\boldsymbol{h}$ is a last-layer feature and $\boldsymbol{\mu}_k, \ \forall \ 1 \leq k \leq K$ are the class means of last-layer features. The proportion of disagreement decreases as training advances. This result indicates that the classifier's decision is gradually simplified to the nearest class mean decision rule.

## C  PROOF FOR THEOREM 1

Note that the constraint constrained optimization problem of Eq. 5 is reduced to an entangled constraint. We consider the $k$-th ($1 \le k \le K$ and $K \ge 2$) problem as:

$$
\min_{H} \quad \frac{1}{N} \sum_{i=1}^{n_k} \mathcal{L}_{ce}\left(\mathbf{h}_{k,i}, \mathbf{Q}^*\right),
$$
$$
s.t. \ \|\mathbf{h}_{k,i}\|^2 - 2\sum_{j \neq k}^{K} \mathbf{h}_{k,i}^{\mathsf{T}}\mathbf{q}_j^* \le E_{HQ}, \ 1 \le i \le n_k,
\tag{24}
$$

where $\mathbf{Q}^*$ is the fixed non-negative orthogonal classifiers. The entangled constraint in Eq. 24 leads to a different conclusion of the proof compared to Theorem 1 in Yang et al. (2022b). This is achieved by inducing non-negativity and orthogonality to the classifier, while also considering the fixed scale of the weight parameters in the classifier. The problem described above is convex, as the objective is a sum of affine functions and log-sum-exp functions with convex constraints.

We have the Lagrange function as:

$$
\tilde{L} = \frac{1}{N} \sum_{i=1}^{n_k} -\log\frac{\exp(\mathbf{h}_{k,i}^{\mathsf{T}}\mathbf{q}_k^*)}{\sum_{j=1}^{K}\exp(\mathbf{h}_{k,i}^{\mathsf{T}}\mathbf{q}_j^*)} + \sum_{i=1}^{n_k}\lambda_i\left(\|\mathbf{h}_{k,i}\|^2 - 2\sum_{j \neq k}^{K}\mathbf{h}_{k,i}^{\mathsf{T}}\mathbf{q}_j^* - E_{HQ}\right),
\tag{25}
$$

where $\lambda_i$ is the Langrange multiplier. We have its gradient with respect to $\mathbf{h}_{k,i}$ as:

$$
\frac{\partial\tilde{L}}{\partial\mathbf{h}_{k,i}} = -\frac{(1-p_k)\,\mathbf{q}_k^*}{N} + \frac{1}{N}\sum_{j \neq k}^{K}p_j\mathbf{q}_j^* + 2\lambda_i\left(\mathbf{h}_{k,i} - \sum_{j \neq k}^{K}\mathbf{q}_j^*\right), 1 \le i \le n_k.
\tag{26}
$$

First we consider the case when $\lambda_i = 0$. $\partial\tilde{L}/\partial\mathbf{h}_{k,i} = 0$ gives the following equation:

$$
\frac{(1-p_k)\,\mathbf{q}_k^*}{N} = \frac{1}{N}\sum_{j \neq k}^{K}p_j\mathbf{q}_j^*
$$
$$
\sum_{j \neq k}^{K}p_j\mathbf{q}_k^* = \sum_{j \neq k}^{K}p_j\mathbf{q}_j^*.
\tag{27}
$$

Multiplying $\mathbf{q}_k^*$ by both sides of the equation, we should have:

$$
\sum_{j \neq k}^{K}p_j = 0 \ \ (\because \mathbf{q}_k^{*\mathsf{T}}\mathbf{q}_{k'}^* = \delta_{k,k'}, \ \forall k,k' \in [1,K]),
\tag{28}
$$

which contradicts with $p_j > 0, \forall 1 \le i \le K$ when the $\ell_2$ norm of $\mathbf{h}_{k,i}$ is constrained and $\mathbf{Q}^*$ has a fixed $\ell_2$ norm. So we have $\lambda_i > 0$ and according to the KKT condition:

$$
\|\mathbf{h}_{k,i}\|^2 - 2\sum_{j \neq k}^{K}\mathbf{h}_{k,i}^{\mathsf{T}}\mathbf{q}_j^* = E_{HQ}.
\tag{29}
$$

Then we have the equation:

$$
\frac{\partial\tilde{L}}{\partial\mathbf{h}_{k,i}^*} = \frac{1}{N}\sum_{j \neq k}^{K}p_j\left(\mathbf{q}_j^* - \mathbf{q}_k^*\right) + 2\lambda_i\left(\mathbf{h}_{k,i}^* - \sum_{j \neq k}^{K}\mathbf{q}_j^*\right) = 0,
\tag{30}
$$

where $\mathbf{h}_{k,i}^*$ is the optimal solution of $\mathbf{h}_{k,i}$. Multiplying $\mathbf{q}_{j'}^*$ ($j' \neq k$) by both sides of Eq. 30, we get:

$$
\frac{p_{j'}}{N} + 2\lambda_i\left(\mathbf{h}_{k,i}^{*\mathsf{T}}\mathbf{q}_j^* - 1\right) = 0.
\tag{31}
$$

Since $p_{j'} > 0$, we have $\mathbf{h}_{k,i}^{*\mathsf{T}}\mathbf{q}_j^* < 1$. Then for any pair $j,j' \neq k$, we have:

$$
\frac{p_j}{p_{j'}} = \frac{\exp\left(\mathbf{h}_{k,i}^{*\mathsf{T}}\mathbf{q}_j^*\right)}{\exp\left(\mathbf{h}_{k,i}^{*\mathsf{T}}\mathbf{q}_{j'}^*\right)} = \frac{\exp\left(\mathbf{h}_{k,i}^{*\mathsf{T}}\mathbf{q}_j^* - 1\right)}{\exp\left(\mathbf{h}_{k,i}^{*\mathsf{T}}\mathbf{q}_{j'}^* - 1\right)} = \frac{\mathbf{h}_{k,i}^{*\mathsf{T}}\mathbf{q}_j^* - 1}{\mathbf{h}_{k,i}^{*\mathsf{T}}\mathbf{q}_{j'}^* - 1}.
\tag{32}
$$

Considering that the function $f(x) = x/\exp(x)$ is monotonically increasing when $x < 0$, we have:

$$\frac{\mathbf{h}_{k,i}^{*\mathsf{T}}\mathbf{q}_j^* - 1}{\exp\left(\mathbf{h}_{k,i}^{*\mathsf{T}}\mathbf{q}_j^* - 1\right)} = \frac{\mathbf{h}_{k,i}^{*\mathsf{T}}\mathbf{q}_{j'}^* - 1}{\exp\left(\mathbf{h}_{k,i}^{*\mathsf{T}}\mathbf{q}_{j'}^* - 1\right)} \tag{33}$$

$$\mathbf{h}_{k,i}^{*\mathsf{T}}\mathbf{q}_j^* = \mathbf{h}_{k,i}^{*\mathsf{T}}\mathbf{q}_{j'}^* = C, p_j = p_{j'} = p, \ \forall j, j' \neq k,$$

where $C$ and $p$ are constants. From Eq. 31, we have:

$$p = -2N\lambda_i(C - 1), \tag{34}$$

and

$$1 - p_k = (K - 1)p = -2N\lambda_i(K - 1)(C - 1). \tag{35}$$

From Eq. 30, we have:

$$
\begin{aligned}
\mathbf{h}_{k,i}^* &= \frac{1}{2N\lambda_i}\left((1 - p_k)\,\mathbf{q}_k^* - \sum_{j \neq k}^{K} p_j\mathbf{q}_j^*\right) + \sum_{j \neq k}^{K}\mathbf{q}_j^* \\
&= \frac{1}{2N\lambda_i}\left(-2N\lambda_i(K-1)(C-1)\mathbf{q}_k^* + 2N\lambda_i(C-1)\sum_{j\neq k}^{K}\mathbf{q}_j^*\right) + \sum_{j\neq k}^{K}\mathbf{q}_j^* \\
&= -(K-1)(C-1)\mathbf{q}_k^* + C\sum_{j\neq k}^{K}\mathbf{q}_j^*,
\end{aligned}
\tag{36}
$$

The classifier in OLPM differs from Yang et al. (2022b) in that it is not a simplex ETF, where the proof is concluded by simplifying Eq. 36. Under the different condition, incorporating the max-margin concept outlined in Theorem 3.1 of Ji et al. (2022) provides an additional theoretical benefit in our proof.

The margin of a single feature $\mathbf{h}_{k,i}^*$ is defined as:

$$\mathcal{M}_{k,i} := \mathbf{h}_{k,i}^{*\mathsf{T}}\mathbf{q}_k^* - \max_{j \neq k}\mathbf{h}_{k,i}^{*\mathsf{T}}\mathbf{q}_j^*. \tag{37}$$

Multiplying $\mathbf{q}_k^*$ by both sides of Eq. 36, we should have:

$$\mathbf{h}_{k,i}^{*\mathsf{T}}\mathbf{q}_k^* = -(K-1)(C-1) \leq K-1 \ \ (\because 0 \leq C < 1), \tag{38}$$

and

$$\mathbf{h}_{k,i}^{*\mathsf{T}}\mathbf{q}_j^* = \mathbf{h}_{k,i}^{*\mathsf{T}}\mathbf{q}_{j'}^* = C \geq 0, \ \ \forall j, j' \neq k \tag{39}$$

When the equality holds, we have:

$$
\begin{aligned}
\mathbf{h}_{k,i}^{*\mathsf{T}}\mathbf{q}_k^* &= K-1 \\
\mathbf{h}_{k,i}^{*\mathsf{T}}\mathbf{q}_j^* &= 0, \ \ \forall j \neq k,
\end{aligned}
\tag{40}
$$

which is equivalent to Eq. 7 and concludes the proof.

# D  ALGORITHMS FOR SECTION 6.1 AND 6.2

In this section, we provide algorithms that are more detailed and may help with implementation for *masked and weighted softmax with the FNO classifier in the task $T_t$* in section 6.1 and for *arc-mixup with the FNO classifier and feature masking in the mini-batch* $\mathbb{B}$ in section 6.2. These algorithms represent the whole process in training classification models in continual learning and imbalanced learning, respectively.

---

**Algorithm 1** Masked and Weighted Softmax with the FNO classifier in the Task $T_t$

---

**Require:** $(\boldsymbol{X}_t, \boldsymbol{Y}_t), \boldsymbol{Q}$
**Ensure:** $\mathbf{P} \in \mathbb{R}^{N_t \times K}$
1: $\mathbf{H} \leftarrow \text{RELU}(f_\theta(\boldsymbol{X}_t))$        ▷ get features from $\boldsymbol{X}_t$ as Eq. 1
2: $\mathbb{K} = \{c_i \mid c_i \text{ of } \boldsymbol{X}_t\}$        ▷ initialize a set of class labels in the task $T_t$.
3: $M^{(-\infty)} = (m_{i,j})_{1 \le i \le N_t, 1 \le j \le K}$ , where $m_{i,j} = \text{-}\infty$ if $j \notin \mathbb{K}$ otherwise 1      ▷ initialize $M^{(-\infty)}$
4: $\mathbf{P} = \text{W-SOFTMAX}(M^{(-\infty)} \odot \text{MATMUL}(\boldsymbol{Q}, \mathbf{H}))$        ▷ get the confidence of $\mathbf{H}$

---

**Algorithm 2** Arc-mixup with the FNO classifier and feature masking in the mini-batch $\mathbb{B}$

---

**Require:** $(\boldsymbol{X}, \boldsymbol{Y}) \in \mathbb{B}, \boldsymbol{Q}$
**Ensure:** $\mathbf{P} \in \mathbb{R}^{|\mathbb{B}| \times K}$
1: $(\hat{\boldsymbol{X}}, \hat{\boldsymbol{Q}}) \leftarrow ArcMixup(\boldsymbol{X}, \boldsymbol{Q})$        ▷ mixup input samples and class vectors as Eq. 11
2: $\hat{\mathbf{H}} \leftarrow \text{RELU}(f_\theta(\hat{\boldsymbol{X}}))$        ▷ get features from $\hat{\boldsymbol{X}}$ as Eq. 1
3: $\mathbb{K} = \{c_i \mid c_i \in \mathbb{B}, \forall 1 \le i \le |\mathbb{B}|\}$        ▷ initialize a set of class labels in the mini-batch
4: $\hat{\mathbb{J}} = \bigcup_{k \in \mathbb{K}} \mathbb{J}_k$        ▷ initialize an index set including all index sets of $\boldsymbol{Q}$
5: $M^{(0)} = (m_{i,j})_{1 \le i \le |\mathbb{B}|, 1 \le j \le D}$ , where $m_{i,j} = \mathbf{1}_{j \in \hat{\mathbb{J}}}$        ▷ initialize a zero mask $M^{(0)}$
6: $\mathbf{P} = \text{MATMUL}(\hat{\boldsymbol{Q}}, \text{LAYERNORM}(M^{(0)} \odot \hat{\mathbf{H}}))$        ▷ get the confidence of $\hat{\mathbf{H}}$

---

# E  ADDITIONAL IMPLEMENTATION DETAILS FOR EXPERIMENTS IN SECTION 7

## E.1  CONTINUAL LEARNING IN SPLIT DATASETS

**Dataset Explanation.** Following Hsu et al. (2018); Van de Ven & Tolias (2019); Buzzega et al. (2020); Kim et al. (2023a), the split datasets of the MNIST, CIFAR10, CIFAR100, and Tiny-ImageNet are described as:

- **S-MNIST, S-CIFAR10**: $T_1(0\text{-}1)$, $T_2(2\text{-}3)$, $T_3(4\text{-}5)$, $T_4(6\text{-}7)$, $T_5(8\text{-}9)$

- **S-CIFAR100**: $T_1(0\text{-}9)$, $T_2(10\text{-}19)$, $T_3(20\text{-}29)$, $T_4(30\text{-}39)$, $T_5(40\text{-}49)$, $T_6(50\text{-}59)$, $T_7(60\text{-}69)$, $T_8(70\text{-}79)$, $T_9(80\text{-}89)$, $T_{10}(90\text{-}99)$

- **S-Tiny-ImageNet**: $T_1(0\text{-}9)$, $T_2(10\text{-}19)$, $T_3(20\text{-}29)$, $T_4(30\text{-}39)$, $T_5(40\text{-}49)$, $T_6(50\text{-}59)$, $T_7(60\text{-}69)$, $T_8(70\text{-}79)$, $T_9(80\text{-}89)$, $T_{10}(90\text{-}99)$ $T_{11}(100\text{-}109)$, $T_{12}(110\text{-}119)$, $T_{13}(120\text{-}129)$, $T_{14}(130\text{-}139)$, $T_{15}(140\text{-}149)$, $T_{16}(150\text{-}159)$, $T_{17}(160\text{-}169)$, $T_{18}(170\text{-}179)$, $T_{19}(180\text{-}189)$, $T_{20}(190\text{-}199)$

where $T_t(i\text{-}j)$ indicates that the $t$-th task has class labels from $i$ to $j$ and all classes are sequentially divided to each task. In CIFAR10, for instance, we can convert the class index in each task as below:

- $T_1(0\text{-}1)$: {airplane, autombile}

- $T_2(2\text{-}3)$: {bird, cat}

- $T_3(4\text{-}5)$: {deer, dog}

- $T_4(6\text{-}7)$: {frog, horse}

- $T_5(8\text{-}9)$: {ship, truck}

Table 6: Hyperparameter settings on all experiments in Continual Learning Benchmarks. (Table credit: (Kim et al., 2023a)

| Method | Buffer | S-MNIST | | | S-CIFAR10 | | | S-CIFAR100 | | | S-Tiny-ImageNet | | |
|---|---|---|---|---|---|---|---|---|---|---|---|---|---|
| SGD | - | $lr$: 0.03 | | | $lr$: 0.1 | | | $lr$: 0.03 | | | $lr$: 0.03 | | |
| ER | 200 | $lr$: 0.01 | | | $lr$: 0.1 | | | $lr$: 0.1 | | | $lr$: 0.1 | | |
| | 500 | $lr$: 0.1 | | | $lr$: 0.1 | | | $lr$: 0.1 | | | $lr$: 0.03 | | |
| | 5120 | $lr$: 0.1 | | | $lr$: 0.1 | | | $lr$: 0.1 | | | $lr$: 0.1 | | |
| DER++ | 200 | $lr$: 0.03 | $\alpha$: 0.2 | $\beta$: 1.0 | $lr$: 0.03 | $\alpha$: 0.1 | $\beta$: 0.5 | $lr$: 0.03 | $\alpha$: 0.1 | $\beta$: 0.5 | $lr$: 0.03 | $\alpha$: 0.1 | $\beta$: 1.0 |
| | 500 | $lr$: 0.03 | $\alpha$: 1.0 | $\beta$: 0.5 | $lr$: 0.03 | $\alpha$: 0.2 | $\beta$: 0.5 | $lr$: 0.03 | $\alpha$: 0.1 | $\beta$: 0.5 | $lr$: 0.03 | $\alpha$: 0.2 | $\beta$: 0.5 |
| | 5120 | $lr$: 0.1 | $\alpha$: 0.2 | $\beta$: 0.5 | $lr$: 0.03 | $\alpha$: 0.1 | $\beta$: 1.0 | $lr$: 0.03 | $\alpha$: 0.1 | $\beta$: 0.5 | $lr$: 0.03 | $\alpha$: 0.1 | $\beta$: 0.5 |

**Hyperparameter Settings.** We have the same settings to Buzzega et al. (2020); Kim et al. (2023a). To summarize them, we borrow the table of hyperparameter settings in Kim et al. (2023a) (Please refer to Table 6).

**Task-Incremental Learning (Task-IL) and Class-Incremental Learning (Class-IL).** As described in Hsu et al. (2018); Van de Ven & Tolias (2019), the sequence of tasks can be modeled in two ways: Task-IL and Class-IL. Both divide datasets and train models for tasks in the same way. The only one difference lies in the evaluation system. To be more specific, Task-IL provides the task information during prediction, so the models classify test samples into the target task's classes. In contrast, Class-IL does not provide task information, so the models must predict the test samples as one of the total classes, regardless of the target task. We trained and validated our models in continual learning benchmarks using these training strategies.

### E.2 IMBALANCED LEARNING IN LONG-TAILED DATASETS

**Dataset Explanation.** Following Zhong et al. (2021) and Zhou et al. (2020), the long-tailed datasets of CIFAR10, CIFAR100, ImageNet (Russakovsky et al., 2015), and Places365 (Zhou et al., 2017) are described as:

- **CIFAR10-LT** comprises ten imbalanced classes, subsampled at an exponentially decreasing rate from the initial class of CIFAR10 as mentioned in (Zhong et al., 2021).
- **CIFAR100-LT** includes one hundred imbalanced classes, constructed in the same way as CIFAR10-LT.
- **ImageNet-LT** is a long-tailed dataset for large-scale object classification, derived from the ImageNet. The sampling is based on Pareto distribution with a power value $\alpha = 5$ and the classes have varying cardinality from 5 to 1,280. Therefore, it contains 115.8K images sorted into 1,000 classes.
- **Places-LT** is an extended version of the large-scale scene classification dataset Places. The classes differ in their cardinality, ranging from 5 to 4,980, and therefore, it contains 184.5K images from 365 classes.

**Architectures.** We utilize a ResNet32 (Zhong et al., 2021) consisting of three residual blocks, with each output dimensions of 16, 32, and 64, respectively, for the CIFAR10-LT dataset. For the CIFAR100-LT dataset, each output dimension is twice that of the CIFIAR10-LT dataset. Unlike the ResNet architecture for ImageNet, the kernel size, stride, and padding are set to 3, 1, and 1, respectively for the first convolutional layer. ResNet50 and 152 are same to (He et al., 2015).

**Hyperparameters Settings.** We employed ResNet32 for CIFAR10/100-LT and trained the model with 128 mini-batches, utilizing SGD with momentum of 0.9 and weight decay of 2e-4, for 200 epochs. The learning rate was warmed up from 0.02 to the initial learning rate in a linear fashion, divided by 0.1 at epochs 160 and 180. For the other datasets, we utilized ResNet50 and 152, trained the models using SGD with momentum of 0.9 and weight decay of 5e-4, and updated the learning rate using a cosine annealing scheduler. Additionally, we differently set mixup alpha of mixup and arc-mixup as the datasets: $\alpha = 1.0$ for both in CIFAR10/100-LT, $\alpha = 0.2$ for mixup and $\alpha = 5.0$ for arc-mixup in ImageNet, and $\alpha = 0.2$ for both in Places-LT.

# F ADDITIONAL EXPERIMENTAL RESULTS.

## F.1 CONTINUAL LEARNING IN SPLIT DATASETS

**Evaluation Metrics. (Kumari et al., 2022)** Final Average Accuracy ($A_T$) and Forgetting ($F_T$) where $a_{j,t}$ denotes the test accuracy on the $t$-task after the model has trained all task up to $j$.

$$A_T = \frac{1}{T}\sum_{t=1}^{T} a_{T,t} \quad F_T = \frac{1}{T-1}\sum_{t=1}^{T} \max_{j\in\{1,...T-1\}}(a_{j,t} - a_{T,t})$$

Table 7: Classification results for standard continual learning benchmarks including the performance of commonly used methods in these benchmarks. Best in bold for each buffer setting. (Final Average Accuracy ↑ (%): $mean_{std}$)

| $\mathcal{B}$ | Method RM | Clf | M | S-MNIST Class-IL | Task-IL | S-CIFAR-10 Class-IL | Task-IL | S-Tiny-ImageNet Class-IL | Task-IL |
|---|---|---|---|---|---|---|---|---|---|
| - | JOINT | | | $95.57_{0.24}$ | $99.51_{0.07}$ | $92.20_{0.15}$ | $98.31_{0.12}$ | $59.99_{0.19}$ | $82.04_{0.10}$ |
| | SGD | | | $19.60_{0.04}$ | $94.94_{2.18}$ | $19.62_{0.05}$ | $61.02_{3.33}$ | $7.92_{0.26}$ | $18.31_{0.68}$ |
| - | oEWC (Schwarz et al., 2018) | | | $20.46_{1.01}$ | $98.39_{0.48}$ | $19.49_{0.12}$ | $68.29_{3.92}$ | $7.58_{0.10}$ | $19.20_{0.31}$ |
| | SI (Zenke et al., 2017) | | | $19.27_{0.30}$ | $96.00_{2.04}$ | $19.48_{0.17}$ | $68.05_{5.91}$ | $6.58_{0.31}$ | $36.32_{0.13}$ |
| | LwF (Li & Hoiem, 2017) | | | $19.62_{0.01}$ | $94.11_{3.01}$ | $19.61_{0.05}$ | $63.29_{2.35}$ | $8.46_{0.22}$ | $15.85_{0.58}$ |
| | PNN (Rusu et al., 2016) | | | - | $99.23_{0.20}$ | - | $95.13_{0.72}$ | - | $67.84_{0.29}$ |
| 200 | ER (Riemer et al., 2018) | | | $80.43_{1.89}$ | $97.86_{0.35}$ | $44.79_{1.86}$ | $91.19_{0.94}$ | $8.49_{0.16}$ | $38.17_{2.00}$ |
| | GEM (Lopez-Paz & Ranzato, 2017) | | | $80.11_{1.54}$ | $97.78_{0.25}$ | $25.54_{0.76}$ | $90.44_{0.94}$ | - | - |
| | A-GEM (Chaudhry et al., 2018) | | | $45.72_{4.26}$ | $98.61_{0.24}$ | $20.04_{0.34}$ | $83.88_{1.49}$ | $8.07_{0.08}$ | $22.77_{0.03}$ |
| | iCaRL (Rebuffi et al., 2017) | | | $70.51_{0.53}$ | $98.28_{0.09}$ | $49.02_{3.20}$ | $88.99_{2.13}$ | $7.53_{0.79}$ | $28.19_{1.47}$ |
| | FDR (Benjamin et al., 2018) | | | $79.43_{3.26}$ | $97.66_{0.18}$ | $30.91_{2.74}$ | $91.01_{0.68}$ | $8.70_{0.19}$ | $40.36_{0.68}$ |
| | GSS (Aljundi et al., 2019) | | | $38.90_{2.49}$ | $95.02_{1.85}$ | $39.07_{5.59}$ | $88.80_{2.89}$ | - | - |
| | HAL (Chaudhry et al., 2021) | | | $84.70_{0.87}$ | $97.96_{0.21}$ | $32.36_{2.70}$ | $82.51_{3.20}$ | - | - |
| | DER (Buzzega et al., 2020) | | | $84.55_{1.64}$ | $98.80_{0.15}$ | $61.93_{1.79}$ | $91.40_{0.92}$ | $11.87_{0.78}$ | $40.22_{0.67}$ |
| | DER++ (Buzzega et al., 2020) | | | $85.61_{1.40}$ | $98.76_{0.28}$ | $64.88_{1.17}$ | $91.92_{0.60}$ | $10.96_{1.17}$ | $40.87_{1.16}$ |
| | ER | FC | ✓[†] | $82.98_{1.03}$ | $98.13_{0.16}$ | $61.75_{6.07}$ | $91.39_{2.13}$ | $15.47_{0.67}$ | $44.11_{0.50}$ |
| | ER | FNO | ✓[†] | $84.26_{1.16}$ | $98.45_{0.19}$ | $63.84_{1.47}$ | $92.03_{0.52}$ | $17.31_{0.74}$ | $44.76_{0.90}$ |
| | DER++ | FC | ✓[†] | $84.45_{0.88}$ | $99.03_{0.09}$ | $66.35_{1.52}$ | $93.17_{0.54}$ | $13.21_{0.56}$ | $49.75_{0.99}$ |
| | DER++ | FNO | ✓[†] | $\mathbf{86.27}_{0.88}$ | $\mathbf{99.11}_{0.08}$ | $\mathbf{67.53}_{1.25}$ | $\mathbf{93.98}_{0.39}$ | $\mathbf{18.44}_{0.94}$ | $\mathbf{53.06}_{0.67}$ |
| 500 | ER (Riemer et al., 2018) | | | $86.12_{1.89}$ | $99.04_{0.18}$ | $57.74_{0.27}$ | $93.61_{0.27}$ | $9.99_{0.29}$ | $48.64_{0.46}$ |
| | GEM (Lopez-Paz & Ranzato, 2017) | | | $85.99_{1.35}$ | $98.71_{0.20}$ | $26.20_{1.26}$ | $92.16_{0.69}$ | - | - |
| | A-GEM (Chaudhry et al., 2018) | | | $46.66_{5.85}$ | $98.93_{0.21}$ | $22.67_{0.57}$ | $89.48_{1.45}$ | $8.06_{0.04}$ | $25.33_{0.49}$ |
| | iCaRL (Rebuffi et al., 2017) | | | $70.10_{1.08}$ | $98.32_{0.07}$ | $47.55_{3.95}$ | $88.22_{2.62}$ | $9.38_{1.53}$ | $31.55_{3.27}$ |
| | FDR (Benjamin et al., 2018) | | | $85.87_{4.04}$ | $97.54_{1.90}$ | $28.71_{3.23}$ | $93.29_{0.59}$ | $10.54_{0.21}$ | $49.88_{0.71}$ |
| | GSS (Aljundi et al., 2019) | | | $49.76_{4.73}$ | $97.71_{0.53}$ | $49.73_{4.78}$ | $91.02_{1.57}$ | - | - |
| | HAL (Chaudhry et al., 2021) | | | $87.21_{0.49}$ | $98.03_{0.22}$ | $41.79_{4.46}$ | $84.54_{2.36}$ | - | - |
| | DER (Buzzega et al., 2020) | | | $90.54_{1.18}$ | $98.84_{0.13}$ | $70.51_{1.67}$ | $93.40_{0.39}$ | $17.75_{1.14}$ | $51.78_{0.88}$ |
| | DER++ (Buzzega et al., 2020) | | | $91.00_{1.49}$ | $98.94_{0.27}$ | $72.70_{1.36}$ | $93.88_{0.50}$ | $19.38_{1.41}$ | $51.91_{0.68}$ |
| | ER | FC | ✓[†] | $89.35_{0.59}$ | $\mathbf{99.20}_{0.16}$ | $70.64_{1.28}$ | $94.22_{0.41}$ | $20.43_{0.38}$ | $53.21_{0.84}$ |
| | ER | FNO | ✓[†] | $\mathbf{89.42}_{0.72}$ | $99.16_{0.17}$ | $71.43_{0.95}$ | $94.38_{0.43}$ | $22.41_{0.57}$ | $52.60_{0.58}$ |
| | DER++ | FC | ✓[†] | $83.10_{1.22}$ | $99.08_{0.09}$ | $71.85_{3.76}$ | $94.28_{1.49}$ | $17.71_{0.58}$ | $59.86_{1.08}$ |
| | DER++ | FNO | ✓[†] | $86.75_{0.75}$ | $99.00_{0.10}$ | $\mathbf{74.77}_{0.66}$ | $\mathbf{95.56}_{0.16}$ | $\mathbf{22.45}_{0.36}$ | $\mathbf{59.87}_{1.91}$ |
| 5120 | ER (Riemer et al., 2018) | | | $93.40_{1.29}$ | $99.33_{0.22}$ | $82.47_{0.52}$ | $96.98_{0.17}$ | $27.40_{0.31}$ | $67.29_{0.23}$ |
| | GEM (Lopez-Paz & Ranzato, 2017) | | | $95.11_{0.87}$ | $99.44_{0.12}$ | $25.26_{3.46}$ | $95.55_{0.02}$ | - | - |
| | A-GEM (Chaudhry et al., 2018) | | | $54.24_{6.49}$ | $98.93_{0.20}$ | $21.99_{2.29}$ | $90.10_{2.09}$ | $7.96_{0.13}$ | $26.22_{0.65}$ |
| | iCaRL (Rebuffi et al., 2017) | | | $70.60_{1.03}$ | $98.32_{0.11}$ | $55.07_{1.55}$ | $92.23_{0.84}$ | $14.08_{1.92}$ | $40.83_{3.11}$ |
| | FDR (Benjamin et al., 2018) | | | $87.47_{3.15}$ | $97.79_{1.33}$ | $19.70_{0.07}$ | $94.32_{0.97}$ | $28.97_{0.41}$ | $68.01_{0.42}$ |
| | GSS (Aljundi et al., 2019) | | | $89.39_{0.75}$ | $98.33_{0.17}$ | $67.27_{4.27}$ | $94.19_{1.15}$ | - | - |
| | HAL (Chaudhry et al., 2021) | | | $89.52_{0.96}$ | $98.35_{0.17}$ | $59.12_{4.41}$ | $88.51_{3.32}$ | - | - |
| | DER (Buzzega et al., 2020) | | | $94.90_{0.57}$ | $99.29_{0.11}$ | $83.81_{0.33}$ | $95.43_{0.33}$ | $36.73_{0.64}$ | $69.50_{0.26}$ |
| | DER++ (Buzzega et al., 2020) | | | $95.30_{1.20}$ | $99.47_{0.07}$ | $85.24_{0.49}$ | $96.12_{0.21}$ | $39.02_{0.97}$ | $69.84_{0.63}$ |
| | ER | FC | ✓[†] | $93.51_{0.60}$ | $99.38_{0.12}$ | $82.63_{1.34}$ | $96.45_{0.27}$ | $35.73_{0.41}$ | $67.50_{0.53}$ |
| | ER | FNO | ✓[†] | $93.98_{0.39}$ | $99.47_{0.09}$ | $82.88_{1.35}$ | $96.79_{0.38}$ | $36.90_{0.41}$ | $66.86_{0.32}$ |
| | DER++ | FC | ✓[†] | $93.75_{0.23}$ | $\mathbf{99.62}_{0.05}$ | $84.71_{0.65}$ | $96.78_{0.16}$ | $34.72_{0.46}$ | $72.40_{0.25}$ |
| | DER++ | FNO | ✓[†] | $\mathbf{94.26}_{0.24}$ | $99.59_{0.05}$ | $\mathbf{85.65}_{0.38}$ | $\mathbf{97.20}_{0.13}$ | $\mathbf{38.95}_{0.71}$ | $\mathbf{72.70}_{0.27}$ |

Table 8: Classification results for standard CL benchmarks. The experiment on S-Tiny-ImageNet was conducted using 5 trials with random seeds while other experiments were conducted using 10 trials with random seeds. Best in bold for each buffer setting. (Final Average Forgetting $\downarrow$ (%): $mean_{std}$)

| $\mathcal{B}$ | RM | Clr | M Class-IL | S-MNIST Task-IL | Class-IL | S-CIFAR-10 Task-IL | |
|---|---|---|---|---|---|---|---|
| - | SGD | | | $99.10_{0.55}$ | $5.15_{2.74}$ | $96.39_{0.12}$ | $46.24_{2.12}$ |
| - | oEWC (Schwarz et al., 2018) | | | $97.79_{1.24}$ | $0.44_{0.16}$ | $91.64_{3.07}$ | $29.33_{3.84}$ |
| | SI (Zenke et al., 2017) | | | $98.89_{0.86}$ | $5.15_{2.74}$ | $95.78_{0.64}$ | $38.76_{0.89}$ |
| | LwF (Li & Hoiem, 2017) | | | $99.30_{0.11}$ | $5.15_{2.74}$ | $96.69_{0.25}$ | $32.56_{0.56}$ |
| | PNN (Rusu et al., 2016) | | | - | $0.00_{0.00}$ | - | $0.00_{0.00}$ |
| 200 | ER (Riemer et al., 2018) | | | $21.36_{2.46}$ | $0.84_{0.41}$ | $61.24_{2.62}$ | $7.08_{0.64}$ |
| | GEM (Lopez-Paz & Ranzato, 2017) | | | $22.32_{2.04}$ | $1.19_{0.38}$ | $82.61_{1.60}$ | $9.27_{2.07}$ |
| | A-GEM (Chaudhry et al., 2018) | | | $66.15_{6.84}$ | $0.96_{0.28}$ | $95.73_{0.20}$ | $16.39_{0.86}$ |
| | iCaRL (Rebuffi et al., 2017) | | | $11.73_{0.73}$ | $0.28_{0.08}$ | $28.72_{0.49}$ | $2.63_{3.48}$ |
| | FDR (Benjamin et al., 2018) | | | $21.15_{4.18}$ | $0.52_{0.18}$ | $86.40_{2.67}$ | $7.36_{0.03}$ |
| | GSS (Aljundi et al., 2019) | | | $74.10_{3.03}$ | $4.30_{2.31}$ | $75.25_{4.07}$ | $8.56_{1.78}$ |
| | HAL (Chaudhry et al., 2021) | | | $14.54_{1.49}$ | $0.53_{0.19}$ | $69.11_{4.21}$ | $12.26_{0.02}$ |
| | DER (Buzzega et al., 2020) | | | $17.66_{2.10}$ | $0.57_{0.18}$ | $40.76_{0.42}$ | $6.57_{0.20}$ |
| | DER++ (Buzzega et al., 2020) | | | $16.27_{1.73}$ | $0.66_{0.28}$ | $32.59_{2.32}$ | $5.16_{0.21}$ |
| | ER | FC | $\checkmark^{\dagger}$ | $10.22_{1.20}$ | $0.69_{0.12}$ | $26.30_{6.18}$ | $6.18_{1.56}$ |
| | ER | FNO | $\checkmark^{\dagger}$ | $9.88_{1.79}$ | $0.50_{0.16}$ | $\mathbf{21.93}_{3.75}$ | $5.91_{0.69}$ |
| | DER++ | FC | $\checkmark^{\dagger}$ | $\mathbf{4.29}_{0.75}$ | $0.32_{0.09}$ | $23.10_{1.46}$ | $4.82_{0.71}$ |
| | DER++ | FNO | $\checkmark^{\dagger}$ | $5.59_{0.76}$ | $\mathbf{0.25}_{0.05}$ | $27.00_{2.20}$ | $\mathbf{3.83}_{0.68}$ |
| 500 | ER (Riemer et al., 2018) | | | $15.97_{2.46}$ | $0.39_{0.20}$ | $45.35_{0.07}$ | $3.54_{0.35}$ |
| | GEM (Lopez-Paz & Ranzato, 2017) | | | $15.57_{1.77}$ | $0.54_{0.15}$ | $74.31_{4.62}$ | $9.12_{0.21}$ |
| | A-GEM (Chaudhry et al., 2018) | | | $65.84_{7.24}$ | $0.64_{0.20}$ | $94.01_{1.16}$ | $14.26_{4.18}$ |
| | iCaRL (Rebuffi et al., 2017) | | | $11.84_{0.73}$ | $0.30_{0.09}$ | $25.71_{1.10}$ | $2.66_{2.47}$ |
| | FDR (Benjamin et al., 2018) | | | $13.90_{5.19}$ | $1.35_{2.40}$ | $85.62_{0.36}$ | $4.80_{0.00}$ |
| | GSS (Aljundi et al., 2019) | | | $60.35_{6.03}$ | $0.89_{0.40}$ | $62.88_{2.67}$ | $7.73_{3.99}$ |
| | HAL (Chaudhry et al., 2021) | | | $9.97_{1.62}$ | $0.35_{0.21}$ | $62.21_{4.34}$ | $5.41_{1.10}$ |
| | DER (Buzzega et al., 2020) | | | $9.58_{1.52}$ | $0.45_{0.13}$ | $26.74_{0.15}$ | $4.56_{0.45}$ |
| | DER++ (Buzzega et al., 2020) | | | $8.85_{1.86}$ | $0.35_{0.15}$ | $22.38_{4.41}$ | $4.66_{1.15}$ |
| | ER | FC | $\checkmark^{\dagger}$ | $7.18_{1.37}$ | $0.39_{0.17}$ | $17.31_{1.58}$ | $3.09_{0.36}$ |
| | ER | FNO | $\checkmark^{\dagger}$ | $6.79_{1.80}$ | $0.46_{0.20}$ | $\mathbf{15.60}_{2.61}$ | $3.20_{0.48}$ |
| | DER++ | FC | $\checkmark^{\dagger}$ | $\mathbf{2.07}_{0.89}$ | $\mathbf{0.23}_{0.09}$ | $16.95_{3.77}$ | $3.03_{0.76}$ |
| | DER++ | FNO | $\checkmark^{\dagger}$ | $3.81_{0.85}$ | $0.85_{0.28}$ | $17.34_{1.35}$ | $\mathbf{1.97}_{0.37}$ |
| 5120 | ER (Riemer et al., 2018) | | | $6.08_{1.84}$ | $0.25_{0.23}$ | $13.99_{0.12}$ | $0.27_{0.06}$ |
| | GEM (Lopez-Paz & Ranzato, 2017) | | | $4.30_{1.16}$ | $0.16_{0.09}$ | $75.27_{4.41}$ | $6.91_{2.33}$ |
| | A-GEM (Chaudhry et al., 2018) | | | $55.10_{10.79}$ | $0.63_{0.21}$ | $84.49_{3.08}$ | $11.36_{1.68}$ |
| | iCaRL (Rebuffi et al., 2017) | | | $11.64_{0.72}$ | $0.26_{0.06}$ | $24.94_{0.14}$ | $1.59_{0.57}$ |
| | FDR (Benjamin et al., 2018) | | | $11.58_{3.97}$ | $0.95_{1.61}$ | $96.64_{0.19}$ | $1.93_{0.48}$ |
| | GSS (Aljundi et al., 2019) | | | $7.90_{1.21}$ | $0.18_{0.11}$ | $58.11_{9.12}$ | $7.71_{2.31}$ |
| | HAL (Chaudhry et al., 2021) | | | $6.55_{1.63}$ | $0.13_{0.07}$ | $27.19_{7.53}$ | $5.21_{0.50}$ |
| | DER (Buzzega et al., 2020) | | | $4.53_{0.83}$ | $0.32_{0.08}$ | $10.12_{0.80}$ | $2.59_{0.08}$ |
| | DER++ (Buzzega et al., 2020) | | | $4.19_{1.63}$ | $0.23_{0.06}$ | $7.27_{0.84}$ | $1.18_{0.19}$ |
| | ER | FC | $\checkmark^{\dagger}$ | $3.12_{1.10}$ | $0.24_{0.10}$ | $7.51_{1.31}$ | $0.81_{0.28}$ |
| | ER | FNO | $\checkmark^{\dagger}$ | $2.44_{0.60}$ | $0.16_{0.13}$ | $6.16_{1.03}$ | $0.56_{0.20}$ |
| | DER++ | FC | $\checkmark^{\dagger}$ | $\mathbf{0.90}_{0.18}$ | $\mathbf{0.09}_{0.06}$ | $\mathbf{5.02}_{0.69}$ | $0.71_{0.27}$ |
| | DER++ | FNO | $\checkmark^{\dagger}$ | $1.47_{0.25}$ | $0.11_{0.04}$ | $5.56_{0.37}$ | $\mathbf{0.43}_{0.13}$ |

## F.2 IMBALANCED LEARNING IN LONG-TAILED DATASETS

We present the top-1 test accuracy for three class divisions: *Head-Many* (more than 100 images), *Medium* (20 to 100 images), and *Tail-Few* (less than 20 images). The imbalance factor $\omega$ in CIFAR10/100-LT datasets means the ratio of the number of a head class $n_{max}$ to the number of a tail class $n_{min}$, *i.e.*, $\omega = n_{max}/n_{min}$.

Table 9: Results of Imbalanced Learning with other methods on CIFAR10/100-LT. Each column of Method indicates the augmentation method, the type of classifier, and the type of loss function, respectively. Best in bold. (Accuracy (%): $mean_{std}$)

| Method | | | Reference | CIFAR10-LT | | | CIFAR100-LT | | |
|---|---|---|---|---|---|---|---|---|---|
| Augmentation | Classifier | Loss | | 100 | 50 | 10 | 100 | 50 | 10 |
| - | FC | CE | (Yang et al., 2022b) | $72.10_{0.30}$ | $77.60_{0.30}$ | $87.40_{0.30}$ | - | - | - |
| - | ETF | CE | (Yang et al., 2022b) | $72.90_{0.30}$ | $79.50_{0.20}$ | $87.20_{0.10}$ | - | - | - |
| - | ETF | DR | (Yang et al., 2022b) | $73.00_{0.20}$ | $78.40_{0.20}$ | $86.90_{0.20}$ | - | - | - |
| mixup | FC | CE | (Yang et al., 2022b) | $73.90_{0.30}$ | $79.30_{0.20}$ | $87.80_{0.10}$ | 43.00 | 48.10 | - |
| B-mixup | FC | CE | (Zhang et al., 2022b) | 78.70 | - | **89.60** | - | - | - |
| mixup | ETF | CE | (Yang et al., 2022b) | $67.00_{0.40}$ | $77.20_{0.30}$ | $87.00_{0.20}$ | - | - | - |
| mixup | ETF | DR | (Yang et al., 2022b) | $76.50_{0.30}$ | $81.00_{0.20}$ | $87.70_{0.20}$ | 45.30 | 50.40 | - |
| - | FC | CE | (reproduced.)[†] | $71.86_{0.65}$ | $77.58_{0.48}$ | $87.42_{0.30}$ | $41.65_{0.41}$ | $46.89_{0.38}$ | $60.48_{0.34}$ |
| - | ETF | DR | (reproduced.)[†] | $71.94_{0.64}$ | $77.63_{0.42}$ | $86.30_{0.31}$ | $42.86_{0.44}$ | $48.09_{0.39}$ | $60.43_{0.32}$ |
| mixup | FC | CE | (reproduced.)[†] | $74.24_{0.44}$ | $80.00_{0.54}$ | $89.08_{0.32}$ | $43.80_{0.42}$ | $49.57_{0.37}$ | **$63.90_{0.33}$** |
| mixup | ETF | DR | (reproduced.)[†] | $75.18_{0.49}$ | $80.17_{0.31}$ | $87.29_{0.23}$ | $45.45_{0.38}$ | $50.67_{0.37}$ | $62.84_{0.38}$ |
| arc-mixup | FNO | CE | (reproduced.)[†] | **$82.59_{0.26}$** | **$85.13_{0.25}$** | $89.50_{0.14}$ | **$49.26_{2.82}$** | **$54.44_{2.32}$** | $63.14_{3.82}$ |

Table 10: Detailed Results of Imbalanced Learning on CIFAR10/100-LT. Best in bold. (Aug: the augmentation method, Clf: the type of classifier, $\mathcal{L}$: the type of loss function) (Accuracy (%): $mean_{std}$)

| | Method | | | CIFAR10-LT | | | | CIFAR100-LT | | | |
|---|---|---|---|---|---|---|---|---|---|---|---|
| | Aug | Clf | $\mathcal{L}$ | Many | Median | Few | All | Many | Median | Few | All |
| imb 100 | - | FC | CE[†] | $91.99_{3.02}$ | $71.72_{1.67}$ | $51.92_{5.34}$ | $71.86_{0.65}$ | $68.41_{0.55}$ | $40.41_{0.75}$ | $9.94_{0.55}$ | $41.65_{0.41}$ |
| | - | ETF | DR[†] | $83.74_{1.84}$ | $71.56_{1.51}$ | $60.62_{3.14}$ | $71.94_{0.64}$ | $67.70_{0.64}$ | $41.89_{0.75}$ | $13.19_{0.72}$ | $42.86_{0.44}$ |
| | mixup | FC | CE[†] | **$94.72_{0.43}$** | $75.70_{1.03}$ | $51.82_{1.61}$ | $74.24_{0.44}$ | **$73.25_{0.52}$** | $42.56_{0.81}$ | $8.73_{0.49}$ | $43.80_{0.42}$ |
| | mixup | ETF | DR[†] | $80.88_{0.91}$ | $74.97_{0.82}$ | $69.75_{1.38}$ | $75.18_{0.49}$ | $70.01_{0.95}$ | $46.82_{0.65}$ | $13.32_{0.63}$ | $45.45_{0.38}$ |
| | arc-mixup | FNO | CE[†] | $84.60_{0.78}$ | **$80.19_{0.71}$** | **$83.79_{0.86}$** | **$82.59_{0.26}$** | $63.46_{3.55}$ | **$53.04_{2.93}$** | **$27.05_{2.08}$** | **$49.26_{2.82}$** |
| imb 50 | - | FC | CE[†] | $93.30_{1.26}$ | $76.74_{1.00}$ | $62.98_{1.91}$ | $77.58_{0.48}$ | $69.95_{0.61}$ | $47.05_{0.41}$ | $18.08_{0.63}$ | $46.89_{0.38}$ |
| | - | ETF | DR[†] | $85.77_{0.96}$ | $76.25_{0.72}$ | $71.34_{1.73}$ | $77.63_{0.42}$ | $69.01_{0.65}$ | $48.75_{0.71}$ | $21.31_{1.01}$ | $48.09_{0.39}$ |
| | mixup | FC | CE[†] | **$94.98_{0.30}$** | $79.98_{0.70}$ | $65.04_{1.58}$ | $80.00_{0.54}$ | **$74.53_{0.42}$** | $50.26_{0.75}$ | $17.75_{0.66}$ | $49.57_{0.37}$ |
| | mixup | ETF | DR[†] | $82.70_{0.62}$ | $78.20_{0.51}$ | $80.28_{0.95}$ | $80.17_{0.31}$ | $70.49_{0.74}$ | $53.26_{0.61}$ | $22.93_{0.90}$ | $50.67_{0.37}$ |
| | arc-mixup | FNO | CE[†] | $86.00_{0.69}$ | **$82.80_{0.56}$** | **$87.38_{0.81}$** | **$85.13_{0.25}$** | $64.38_{2.97}$ | **$57.81_{1.77}$** | **$38.01_{2.49}$** | **$54.44_{2.32}$** |
| imb 10 | - | FC | CE[†] | $94.00_{0.40}$ | $84.77_{0.45}$ | $84.37_{0.64}$ | $87.42_{0.30}$ | $72.53_{0.37}$ | $61.12_{0.53}$ | $44.73_{0.76}$ | $60.48_{0.34}$ |
| | - | ETF | DR[†] | $89.44_{0.92}$ | $83.53_{0.43}$ | $86.83_{1.04}$ | $86.30_{0.31}$ | $71.44_{0.61}$ | $61.87_{0.55}$ | $45.01_{0.84}$ | $60.43_{0.32}$ |
| | mixup | FC | CE[†] | **$95.37_{0.29}$** | **$86.81_{0.47}$** | $85.80_{0.89}$ | $89.08_{0.32}$ | **$76.89_{0.43}$** | $64.65_{0.45}$ | $46.88_{0.68}$ | **$63.90_{0.33}$** |
| | mixup | ETF | DR[†] | $88.46_{0.56}$ | $83.98_{0.47}$ | $90.54_{0.55}$ | $87.29_{0.23}$ | $72.08_{0.51}$ | **$64.90_{0.50}$** | $48.87_{0.79}$ | $62.84_{0.38}$ |
| | arc-mixup | FNO | CE[†] | $89.94_{0.55}$ | $86.59_{0.53}$ | **$92.95_{0.38}$** | **$89.50_{0.14}$** | $66.26_{5.32}$ | $64.70_{3.06}$ | **$57.39_{3.07}$** | $63.14_{3.82}$ |

Table 11: Detailed Results of Imbalanced Learning on Places-LT. All experiments were conducted using 3 trials with random seeds. Best in bold. (Aug: the augmentation method, Clf: the type of classifier, $\mathcal{L}$: the type of loss function) (Accuracy (%): $mean_{std}$)

| Method | | | ResNet152 | | | | ResNet152 (FT) | | | |
|---|---|---|---|---|---|---|---|---|---|---|
| Aug | Clf | $\mathcal{L}$ | Many | Median | Few | All | Many | Median | Few | All |
| - | FC | CE[†] | $40.63_{0.21}$ | $17.69_{0.64}$ | $2.04_{0.44}$ | $22.62_{0.29}$ | $40.77_{0.08}$ | $19.99_{0.67}$ | $4.41_{0.55}$ | $24.16_{0.41}$ |
| - | ETF | DR[†] | $37.51_{0.68}$ | $19.92_{0.30}$ | $2.01_{0.32}$ | $22.45_{0.39}$ | $39.68_{0.10}$ | $22.53_{0.14}$ | $2.47_{0.55}$ | $24.45_{0.18}$ |
| mixup | FC | CE[†] | **$42.10_{0.31}$** | $15.82_{0.60}$ | $0.86_{0.18}$ | $22.10_{0.27}$ | $43.14_{0.63}$ | $20.10_{1.61}$ | $3.32_{1.33}$ | $24.83_{1.19}$ |
| mixup | ETF | DR[†] | $39.04_{0.28}$ | $20.73_{0.14}$ | $0.86_{0.24}$ | $23.11_{0.16}$ | $41.61_{0.47}$ | $23.37_{0.18}$ | $2.50_{0.31}$ | $25.51_{0.08}$ |
| arc-mixup | FNO | CE[†] | $35.82_{0.32}$ | **$31.74_{0.64}$** | **$16.89_{0.23}$** | **$30.07_{0.40}$** | $40.31_{0.18}$ | **$35.31_{0.19}$** | **$20.89_{0.33}$** | **$34.06_{0.10}$** |

