# OpenReview forum: "Fixed Non-negative Orthogonal Classifier: Inducing Zero-mean Neural Collapse with Feature Dimension Separation"
_ICLR.cc/2024/Conference — ICLR 2024 poster_

### Official Review · Reviewer_iab3 · 2023-10-30

**Soundness:** 3 good
**Presentation:** 3 good
**Contribution:** 3 good
**Rating:** 5
**Confidence:** 4

**Summary:**

The article introduces the concept of a "Fixed Non-negative Orthogonal Classifier" and its relationship with the phenomenon of "Zero-mean Neural Collapse." Fixed classifiers in neural networks have shown cost efficiency and even surpassed learnable classifiers in certain benchmarks when incorporating orthogonality. However, the dynamics of fixed orthogonal classifiers concerning neural collapse, where last-layer features converge to a specific form called simplex ETF during training, have not been deeply explored. This paper addresses this gap by introducing the concept of zero-mean neural collapse in non-negative Euclidean space. The authors propose a fixed non-negative orthogonal classifier that optimally induces this collapse, maximizing the margin of an orthogonal layer-peeled model. This classifier also offers advantages in continual learning and imbalanced learning by separating the last-layer feature dimensions. The paper provides comprehensive experiments to validate its claims, demonstrating significant performance improvements.

**Strengths:**

+ 1. The article is well-structured, logically sound, and skillfully written.
+ 2. This paper conduct extensive experiments to justify *zero-mean neural collapse*, which combines the orthogonality and neural collapse.

**Weaknesses:**

The problem that I'm concerned about most is **unclear motivation**. Authors mentioned in the introduction:
*However, neural collapse differently occurs in the fixed orthogonal classifier due to their limitations from geometrical feature: orthogonality.* So, I have two questions, authors should provide more discussions to demonstrate the meaning in the main text:
  + Why do we have to fix classifier as an orthogonal matrix ?
  + Why studying neural collapse with fixed orthogonal classifier is necessary ?

**Questions:**

Does Remark.1 claim that zero-mean neural collapse can achieve max-margin? Consider the binary class classication, the max-margin feature should be Digon, which has the larger angle (180 degrees) than orthogonality (90 degrees).

By the way, the case that D > K is interesting. Authors can refer to [1] and [2].

[1] https://en.wikipedia.org/wiki/Thomson_problem

[2] https://arxiv.org/abs/2310.05351

---

> ### Author Response · Authors · 2023-11-16
> **Responses for weaknesses**
>
> **Weakness 1.** Why do we have to fix classifier as an orthogonal matrix?
> > The reason why fixed classifiers have non-negativity and orthogonality constraints is that if those constraints do not exist in the LPM with a fixed classifier (not a simplex), it is not guaranteed that the collapse phenomenon between class means and their respective class weight vectors occurs when the LPM has the optimal solution.
> However, when the fixed classifier becomes a non-negative and orthogonal matrix, the LPM obtains the optimal solution even in inducing max margin in decision while the collapse occurs.
>
> ---
>
> **Weakness 2.** Why studying neural collapse with fixed orthogonal classifier is necessary?
> > After introducing the necessary and theoretical benefits (optimal solution and max-margin in decision) of non-negativity and orthogonality in the LPM with a fixed classifier, we have proven the utility of those constraints, and in order to explain the collapse phenomenon that occurs when the LPM with a fixed non-negative orthogonal classifier achieves the optimal solution, it becomes necessary to define special neural collapse properties: zero-mean neural collapse.
>
> ---
>
> **Question.** Does Remark.1 claim that zero-mean neural collapse can achieve max-margin? Consider the binary class classication, the max-margin feature should be Digon, which has the larger angle (180 degrees) than orthogonality (90 degrees).
> > According to the constraints of Eq. 5 in the revised paper, last-layer features are located on the non-negative Euclidean space because they followed the ReLU activation function and were centered at the origin.
> In this situation, the max-margin in decision becomes orthogonal.
>
> ---
>
> **Additional References.** By the way, the case that D > K is interesting. Authors can refer to [1] and [2].
> > Thanks for your thoughtful references about the future work.
> We reflexted them to the future work paragraph in Section 8 (Conclusion and Future Work) on page 9 of the revised paper.
> We don't doubt that these materials will be helpful to resolve the limitation.

---

### Official Review · Reviewer_jWfm · 2023-11-01

**Soundness:** 2 fair
**Presentation:** 3 good
**Contribution:** 3 good
**Rating:** 6
**Confidence:** 3

**Summary:**

The study delves into the intricacies of "Fixed Non-negative Orthogonal Classifiers" in the realm of neural networks, emphasizing their potential in inducing "Zero-mean Neural Collapse." While fixed classifiers have historically demonstrated cost-effectiveness and even outperformed learnable ones with orthogonality, their behavior in the context of neural collapse—a phenomenon where last-layer features align to a specific form, the simplex ETF—remains underexplored. Addressing this, the paper pioneers the idea of zero-mean neural collapse within a non-negative Euclidean space and presents a novel classifier that optimally triggers this collapse. This innovation not only maximizes the margin of an orthogonal layer-peeled model but also enhances performance in continual and imbalanced learning scenarios. Through rigorous experimentation, the authors substantiate their findings, showcasing marked performance enhancements.

**Strengths:**

1. Effectiveness: The proposed methods improve the performance in long-tailed learning.
2. Clarity: Overall, the paper is well-written and easy to follow. Besides, the main theoretical result (Theorem 1) is clear and correct.

**Weaknesses:**

1. My main concern is the necessity of the new theory. The main result (Theorem 1) shares a similar formulation with Lemma 4.1 in [1], showing the zero-mean is unnecessary to achieve the neural collapse. Please provide more evidence of the advantages.
2. Although the proposed methods are effective, the connections with the theoretical analysis seem unclear.
3. The orthogonality is accessible when $d \leq K$, could you please discuss the condition $d > K$?
4. It will be more convincing if more competitors on ImageNet-LT and Places-LT are provided.


Ref:

[1] Gao, P., Xu, Q., Wen, P., Yang, Z., Shao, H. and Huang, Q. Feature Directions Matter: Long-Tailed Learning via Rotated Balanced Representation. ICML, 2023.

**Questions:**

Please refer to the weaknesses.

---

> ### Author Response · Authors · 2023-11-16
> **Responses for weaknesses**
>
> **Weakness 1.** My main concern is the necessity of the new theory. The main result (Theorem 1) shares a similar formulation with Lemma 4.1 in [1], showing the zero-mean is unnecessary to achieve the neural collapse. Please provide more evidence of the advantages.
>
> [1] Gao, P., Xu, Q., Wen, P., Yang, Z., Shao, H. and Huang, Q. Feature Directions Matter: Long-Tailed Learning via Rotated Balanced Representation. ICML, 2023.
>
> >**(Detailed explanation of our motivation)** Before describing the answer to [1], we want to emphasize our motivation.
> The reason why fixed classifiers have non-negativity and orthogonality constraints is that if those constraints do not exist in the LPM with a fixed classifier (not a simplex), it is not guaranteed that the collapse phenomenon between class means and their respective class weight vectors occurs when the LPM has the optimal solution.
> However, when the fixed classifier becomes a non-negative and orthogonal matrix, LPM obtains the optimal solution even in inducing max margin in decision while the collapse occurs.
> Then, we have proven the utility of those constraints, and in order to explain the collapse phenomenon that occurs when LPM with a fixed non-negative orthogonal classifier achieves the optimal solution, it becomes necessary to define special neural collapse properties: zero-mean neural collapse.
>
> >**(Main answer)** Lemma 4.1 in [1] indicates the summation of vectors centered to their global mean.
> Therefore, it is a natural result that the average of centered vectors becomes zero as illustrated in the proof of Lemma 4.1 (Appendix B in [1]; p13).
> In contrast, we centered the class means at the origin, which is the same effect that the global mean of class means is considered as zero-mean.
> We defined the collapse properties in this case as zero-mean neural collapse.
> Apart from this, [1] also proposed an orthogonal-constrained classifier that converges to a $K$-equiangular ETF.
> However, this classifier was not analyzed with LPM and did not take into account non-negativity, which is an important factor that causes feature dimension separation in the weight matrix of the fixed orthogonal classifier.
>
> *We added this content to our reference list.
> Please refer to the Section 2 (Related Work) on page 2 of the revised paper.*
>
> ---
>
> **Weakness 2.** Although the proposed methods are effective, the connections with the theoretical analysis seem unclear.
> > To improve the connections, we changed the order of paragraphs in Section 5 on page 5 of the revised paper according to the modified logical flow of detailed explanation of our motivation in W1.
> Additionally, we divided our previous Figure 1 by splitting it into three separate figures (Figure 1, 2, and 3).
> Figure 1 now illustrates the main motivation of our work, which is *how the collapse between class means and class weight vectors occurs in fixed classifier?*.
> Figure 2 describes the our method, FNO classifier.
> Figure 3 provides the utilization of feature dimension separation in continual learning and imbalanced learning.
> We highlight the impact of the FNO classifier, which enhances masked softmax for continual learning and adjusts the original mixup on the hypersphere by using *arc-mixup* for imbalanced learning.
> Please refer to Figure 1, 2, and 3 on pages 1, 5, and 6 of the revised paper for a clear visualization, respectively.
>
> ---
>
> **Weakness 3.** The orthogonality is accessible when $d \leq K$, could you please discuss the condition $d > K$?
> > We guess that this comment is the limitation that $D \geq K$ in Section 8 (Future Work).
> To reduce the confusion, we clarified it as: *$D$ should be greater than or equal to $K$*.
> When $D < K$, the weight matrix $W \in R^{D \times K}$ is impossible to be orthogonal because at least one weight vector $W_i \in R^{D}$ is not linearly independent, *i.e.,* $\exists_j W_j^{T} W_i \neq 0$.
>
> ---
>
> **Weakness 4.** It will be more convincing if more competitors on ImageNet-LT and Places-LT are provided.
> > We borrowed the performance of ETF in ImageNet-LT from [A1].
> Additionally, to ensure a fair comparison, we reproduced the ETF classifier with the same hyperparameter settings to ours for CIFAR10/100-LT, ImageNet-LT and Places-LT.
> Please refer to Tables 3 and 4 on page 9 and Tables 9, 10, and 11 on page 28 of the revised paper.
>
> [A1] Yibo Yang, Liang Xie, Shixiang Chen, Xiangtai Li, Zhouchen Lin, and Dacheng Tao. Do we really
> need a learnable classifier at the end of deep neural network? arXiv preprint arXiv:2203.09081,
> 2022b.

---

> > ### Comment · Reviewer_jWfm · 2023-12-04
> >
> > Thanks for the authors' responses. The revision explains how the theory motivates the proposed method and provides the limitation when $D < K$. To sum up, the main concerns are well-addressed, and I would like to increase my rating.

---

### Official Review · Reviewer_EhX5 · 2023-11-02

**Soundness:** 3 good
**Presentation:** 3 good
**Contribution:** 3 good
**Rating:** 6
**Confidence:** 3

**Summary:**

This paper introduces the Fixed Non-negative Orthogonal Classifier (FNO), which is a novel approach to address the issue of neural collapse in training classification models. The authors propose the concept of zero-mean neural collapse, where the class means are centered at the origin instead of their global mean. The paper empirically validates the effectiveness of these methods in tasks such as continual learning and imbalanced learning.

**Strengths:**

- The paper introduces a novel Fixed Non-negative Orthogonal Classifier (FNO classifier) and proposes the concept of zero-mean neural collapse. This combination of ideas is interesting.

- The paper provides theoretical analysis of the FNO classifier and proves its benefits in terms of inducing zero-mean neural collapse. The experimental results demonstrate the effectiveness of the FNO classifier in both continual learning and imbalanced learning scenarios.

- The paper is well-structured and clearly explains the motivation, methodology, and results

**Weaknesses:**

- The experiments in the paper are limited to continual and imbalanced learning scenarios for the FNO classifier. It would be beneficial to see how the FNO classifier performs compared to the ETF classifier in standard classification tasks. Additionally, in Table 4, which details the imbalanced learning experiments, the ETF classifier is absent from the comparison. Including it could provide a more comprehensive evaluation of the FNO classifier's performance.

- A related work is missing for discussion. The orthogonality of the classifier in NC is explored funder MSE loss:

Zhou, Jinxin, et al. "On the optimization landscape of neural collapse under mse loss: Global optimality with unconstrained features." International Conference on Machine Learning. PMLR, 2022.

For a full comparison, we may also want to consider the incorporation of MSE loss within the ETF classifier, which is guaranteed to be orthogonal classifier (assuming no bias)

**Questions:**

See the weaknesses part above

---

> ### Author Response · Authors · 2023-11-16
> **Responses for weaknesses**
>
> **Weakness 1-1.** The experiments in the paper are limited to continual and imbalanced learning scenarios for the FNO classifier. It would be beneficial to see how the FNO classifier performs compared to the ETF classifier in standard classification tasks.
>
> |Architecture|Method|MNIST|Fashion MNIST|CIFAR10|CIFAR100|
> |----|----|----|----|----|----|
> |ResNet|FC|99.64|93.56|89.45|65.11|
> |ResNet|ETF|99.66|93.82|88.05|64.23|
> |ResNet|FNO|99.67|94.28|89.59|66.20|
> > We conducted experiments of ETF in standard classification tasks.
> For a fair comparison, we used the same environmental settings to ours when utilizing the analyses of ZNC in Appendix B.1 (Zero-Mean Neural Collapse in Image Classification Benchmarks) on page 20 of the revised paper.
> As shown the above table, our FNO outperforms ETF in standard classification tasks.
>
> **Weakness 1-2.** Additionally, in Table 4, which details the imbalanced learning experiments, the ETF classifier is absent from the comparison. Including it could provide a more comprehensive evaluation of the FNO classifier's performance.
> > We borrowed the performance of ETF in ImageNet-LT from [A1] and also reproduced them in the same environmental setup with ours for fair comparison.
> Including the additional results, our FNO classifier with arc-mixup has consistently outperformed.
> Please refer to Table 4 on page 9 of the revised paper.
>
> ---
>
> **Weakness 2-1.** A related work is missing for discussion. The orthogonality of the classifier in NC is explored funder MSE loss
> > The findings of neural collapse under MSE loss [A2] yield similar results to ours.
> Specifically, the class means and class weight vectors collapse in orthogonal shape.
> However, their classifier was not fixed and the cross entropy loss, which is the most widely used loss function in classification models, was not utilized.
> These differences highlight the improved utility of our work.
> We have included this content in our reference list.
> Please refer to the end of Neural collapse and Orthogonality paragraph in the related work on page 3 of the revised paper.
>
> **Weakness 2-2.** For a full comparison, we may also want to consider the incorporation of MSE loss within the ETF classifier, which is guaranteed to be orthogonal classifier (assuming no bias)
> > When incorporating MSE loss within the fixed ETF classifier, the shape where trained class means converge will vary while aligning to their respective class weight vectors in a similar way to [A1], *i.e.*, it is not guaranteed that the class means collapse their respective class weight vectors at the orthogonal matrix when the fixed classifier is not orthogonal.
>
> ---
>
> [A1] Yibo Yang, Liang Xie, Shixiang Chen, Xiangtai Li, Zhouchen Lin, and Dacheng Tao. Do we really
> need a learnable classifier at the end of deep neural network? arXiv preprint arXiv:2203.09081,
> 2022b.
>
> [A2] Zhou, Jinxin, et al. "On the optimization landscape of neural collapse under mse loss: Global optimality with unconstrained features." International Conference on Machine Learning. PMLR, 2022.

---

### Official Review · Reviewer_WNdB · 2023-11-02

**Soundness:** 2 fair
**Presentation:** 2 fair
**Contribution:** 2 fair
**Rating:** 6
**Confidence:** 4

**Summary:**

The paper delves into the phenomenon of neural collapse, specifically in scenarios with fixed classifiers composed of orthogonal class prototypes. A central assertion of the paper is that neural collapse manifests differently when the classifier is fixed. To address this, the concept of 'zero-mean neural collapse' is introduced. This approach redefines neural collapse by centering class means to the origin in non-negative Euclidean space, rather than to their global mean. The occurrence of Zero-mean Neural Collapse (ZNC) is observed when the orthogonal Layer Peeled Model (LPM) achieves global optimality, simultaneously inducing a max-margin in decision-making. The paper further explores the implications of this phenomenon in the contexts of continual learning and imbalanced learning.

**Strengths:**

The paper poses an interesting problem and a good methodological choice for the substantiation of its main intentions. The work includes a comprehensive part of experiments across diverse contexts and with datasets the introduction and the related works are rich of interesting information and insights.

**Weaknesses:**

The manuscript's writing and structure require refinement for better clarity and flow.
The concepts of masking in continual learning and mixup in imbalanced learning emerge unexpectedly within the text and would benefit from a better introduction with an improved link with neural collapse.

The introduction and related work sections could be condensed to allow for a more comprehensive introduction of Section 6.

The significance of Zero-mean Neural Collapse (ZNC) in non-negative Euclidean space (i.e., the positive hyper-octant) is not immediately apparent. The paper should clarify whether its importance is solely due to the optimality shown in the LPM model or if there are additional factors which are outside the proof. The rationale behind constraining the representation space to the positive hyper-octant warrants further explanation.

The nature of the problem posed by the LPM model is not shown. The manuscript should specify whether it is linear, non-linear, or solvable by known matrix factorization techniques. Moreover, the discussion on the complexity of providing values for W is insufficiently developed, leaving the reader questioning where the complexity of the problem truly lies.

Could the authors provide insight into why LPM optimality does not manifest in the case of a regular fixed d-simplex, and conversely, why it appears to be present in the context of Zero-mean Neural Collapse (ZNC)?

The visual clarity and structural coherence of Figure 1 could be enhanced to better convey the intended information.

The tables detailing experimental results should more clearly differentiate the methodologies used, to avoid confusion. The complex nomenclature, such as FNODERMR++, could be simplified for better clarity.

In Remark 1 at the end of section 5, the statement regarding the inability of a fixed orthogonal classifier to address neural collapse needs further clarification. A more detailed explanation could help in understanding this assertion.

**Questions:**

Weaknesses and questions are grouped above to assist in the association and subsequent discussion of the issues.

---

> ### Author Response · Authors · 2023-11-16
> **Responses for weaknesses 1-3**
>
> **Weakness 1-1.** The manuscript's writing and structure require refinement for better clarity and flow.
> > To clarify our motivation, we highlight that "the collapse does not occur at a simplex ETF in the fixed classifier when the shape of weight matrix in it is not a simplex ETF".
> Additionally, we included more specific visual materials for better comprehension in Figure 1 on page 1 of the revised paper.
>
> **Weakness 1-2.** The concepts of masking in continual learning and mixup in imbalanced learning emerge unexpectedly within the text and would benefit from a better introduction with an improved link with neural collapse.
> > To enhance the coherence of our paper's structure, we more specifically introduce the masking concept in continual learning and arc-mixup adjusted on the hypersphere in imbalanced learning:
>  "In addition, the FNO classifier becomes linearly independent due to non-negativity and orthogonality.
> As a result, some elements in the last-layer feature engaged to one class will not be able to affect other classes.
> This *feature dimension separation* (FDS) synergizes with masked softmax by decreasing interferences between classes in continual learning and enables *arc-mixup* by adjusting the mixup strategy on the hypersphere to work correctly in imbalanced learning."
>
> *All modifications were reflected in Section 1 (Introduction) on page 2 of the revised paper.
> We anticipate that these additions will facilitate readers' comprehension of our work.*
>
> ---
>
> **Weakness 2.** The introduction and related work sections could be condensed to allow for a more comprehensive introduction of Section 6.
> > In addition to W1's, we have revised the Section 1 (Introduction) and Section 2 (Related Work) to provide a concise but specific overview.
> For more comprehensive introduction to Section 6, we provide additional explanations to the flow from zero-mean neural collapse, our FNO classifier, and the benefits in continual learning and imbalanced learning at the end of the Section 2 on page 3 of the revised paper:
> "Based on the different geometric feature of orthogonality, we have developed methods with the conviction that a fixed orthogonal classifier has potential in both continual learning and imbalanced learning much like a fixed simplex ETF, despite it not converging to a simplex ETF.
> With the intuition, we examined the collapse phenomenon that occurs when training a classification model with the fixed orthogonal classifier in the non-negative Euclidean space during TPT, which we have termed *zero-mean neural collapse*.
> We then proposed a *fixed non-negative orthogonal classifier* that satisfies the properties of zero-mean neural collapse while also providing benefits in continual learning and imbalanced learning by class-wise separation of the last-layer feature dimensions."
>
> ---
>
> **Weakness 3-1.** The significance of Zero-mean Neural Collapse (ZNC) in non-negative Euclidean space (i.e., the positive hyper-octant) is not immediately apparent. The paper should clarify whether its importance is solely due to the optimality shown in the LPM model or if there are additional factors which are outside the proof.
> > **(Clarify ZNC's importance in the LPM model.)** The reason why fixed classifiers have non-negativity and orthogonality constraints is that if those constraints do not exist in LPM with a fixed classifier (not a simplex), it is not guaranteed that the collapse phenomenon between class means and their respective class weight vectors occurs when LPM has the optimal solution.
> However, when the fixed classifier becomes a non-negative and orthogonal matrix, LPM obtains the optimal solution even in inducing max margin in decision while the collapse occurs.
> Then, we have proven the utility of those constraints, and in order to explain the collapse phenomenon that occurs when LPM with a fixed non-negative orthogonal classifier achieves the optimal solution, it becomes necessary to define special neural collapse properties: zero-mean neural collapse.
>
> > **(Additional factors.)** Upon the constraints of ZNC, non-negativity and orthogonality, we propose the fixed non-negative orthogonal classifier, figure out their additional benefit (called *feature dimension split*), and utilize our method with masked softmax in continual learning and arc-mixup in imbalanced learning.
>
> **Weakness 3-2.** The rationale behind constraining the representation space to the positive hyper-octant warrants further explanation.
> > Non-negativity and orthogonality are highly beneficial for LPM with a fixed classifier to have the optimality while inducing the collapse.
> As a result, all class weight vectors are restricted on the positive hyper-octant.
> In addition, the last-layer features follow the ReLU and are centered at the origin.
> Therefore, all weights and features are located on the positive hyper-octant.
>
> *Following the above logical flow, we properly changed the order of paragraphs in Section 5 on page 5 of the revised paper.*
>
> ---

---

> ### Author Response · Authors · 2023-11-16
> **Responses for weaknesses 4-7**
>
> **Weakness 4-1.** The nature of the problem posed by the LPM model is not shown. The manuscript should specify whether it is linear, non-linear, or solvable by known matrix factorization techniques.
> > [A1] introduced Layer-Peeled Model (LPM), for Eq. 4 *(on page 3 of our revised paper)* with the goal of unveiling quantitative patterns of deep neural networks.
> The last-layer features $H$ comes from the previous $L-1$ layers which have a non-linear activation function such as the ReLU, so LPM in the new optimization program is non-linear.
>
> **Weakness 4-2.** Moreover, the discussion on the complexity of providing values for W is insufficiently developed, leaving the reader questioning where the complexity of the problem truly lies.
> > **(In terms of general LPM model)** [A1] introduced Layer-Peeled Model (LPM), for Eq. 4 *(on page 3 of our revised paper)* with the goal of unveiling quantitative patterns of deep neural networks.
> This new optimization program remains non-convex  but is likely much more conductive to analysis than the deep neural networks by considering only last-layer features and classification layer.
>
> > **(In terms of our LPM model)** The primary objective of LPM with a fixed classifier (not a simplex) is to observe how the collapse happens when the LPM has an optimal solution.
> In this case, it is not assured that the class means and class weight vectors will collapse in a simplex ETF.
> As a result, our main challenge in the LPM with a fixed classifier becomes to identify the appropriate constraints for the LPM to have the optimal solution while the collapse occurs, not to optimize the optimal solution of $W$ (because it is fixed).
>
> [A1] Cong Fang, Hangfeng He, Qi Long, and Weijie J. Su. Exploring deep neural networks via layer-peeled-model: Minority collapse in imbalanced training. Proceedings of the National Academy of Sciences, 118(43), oct 2021.
>
> ---
>
> **Weakness 5.** Could the authors provide insight into why LPM optimality does not manifest in the case of a regular fixed d-simplex, and conversely, why it appears to be present in the context of Zero-mean Neural Collapse (ZNC)?
> > First of all, it should be noted that LPM with a regular fixed d-simplex has the optimal solution when neural collapse occurs [A2].
> In contrast, we examined the scenario where the shape of a fixed classifier in LPM is not a simplex ETF.
> To the best of our knowledge, it is not guaranteed that the class means of LPM with unconstrained weights it is not assured that the class means converge to a simplex ETF and collapse to class weight vectors
> Therefore, a new method is necessary to invoke the collapse when the shape of the fixed classifier is not a simplex.
> Finally, we discovered that non-negativity and orthogonality are beneficial constraints for the collapse and proved that LPM with a fixed non-negative orthogonal classifier has the optimal solution and max-margin in decision simultaneously when the collapse deviates from usual.
> To clarify this different collapse, a specific set of properties is required.
> As a result, we present a zero-mean neural collapse where the global mean of class means is considered as the origin.
>
> [A2] Yibo Yang, Liang Xie, Shixiang Chen, Xiangtai Li, Zhouchen Lin, and Dacheng Tao. Do we really
> need a learnable classifier at the end of deep neural network? arXiv preprint arXiv:2203.09081,
> 2022b.
>
> ---
>
> **Weakness 6.** The visual clarity and structural coherence of Figure 1 could be enhanced to better convey the intended information.
> > We have revised our previous Figure 1 by splitting it into three separate figures (Figure 1, 2, and 3).
> Figure 1 depicts how the collapse between class means and class weight vectors occurs in fixed classifier.
> Figure 2 describes the our method, FNO classifier.
> Figure 3 provides an overview of two applications of the FNO classifier.
> We highlight the impact of the FNO classifier, which enhances masked softmax for continual learning and adjusts the original mixup on the hypersphere by using *arc-mixup* for imbalanced learning.
> Please refer to Figure 1, 2, and 3 on pages 1, 5, and 6 of the revised paper for a clear visualization, respectively.
>
> ---
>
> **Weakness 7.** The tables detailing experimental results should more clearly differentiate the methodologies used, to avoid confusion. The complex nomenclature, such as FNODERMR++, could be simplified for better clarity.
> > We divided the column of Method in original Table 2 into three parts: RM (indicating rehearsal-based method used), Clf (indicating classifier used), M (indicating whether a negative infinite mask was applied or not).
> The abbreviations FC and FNO indicate a learnable classifier and our FNO classifier, respectively.
> All modifications were reflected and please refer to Table 2 on page 8 of the revised paper.
>
> ---

---

> ### Author Response · Authors · 2023-11-16
> **Response for weakness 8**
>
> **Weakness 8.** In Remark 1 at the end of section 5, the statement regarding the inability of a fixed orthogonal classifier to address neural collapse needs further clarification. A more detailed explanation could help in understanding this assertion.
> > As illustrated in Figure 1 of the revised paper, when the weights of classifier are fixed in a different shape (not a simplex ETF), trained class means does not converge to a simplex ETF form and the collapse to their respective class weight vectors is not assured.
> We explain it briefly at the end of the 2nd paragraph in Section 1 (introduction) on page 1 and specifically at the start of Section 5 (Fixed Non-negative Orthogonal Classifier) on page 5 of the revised paper.

---

### Author Response · Authors · 2023-11-16
**Official comment from authors**

We appreciate all the valuable and helpful comments.  We respond to each comment with an answer or revision to reduce the concerns. We hope that all these responses are considered for re-evaluation of our paper. In particular, the revision about the weakness in delivering clear motivation did not add any additional research materials to the initial manuscript, except explanation and reproduced experimental results for the revision.

---

### Meta-Review · Area_Chair_8HFB · 2023-12-09

**Metareview:**

This study investigates the training dynamics of neural collapse when using a fixed orthogonal classifier. A fixed non-negative orthogonal classifier is proposed with theoretical benefits. The effectiveness is verified in continual learning and imbalance learning with arc-mixup. Multiple reviewers recognize that the idea of combining orthogonality and neural collapse is interesting. The paper is well-organized and easy to follow. The ACs agree with the reviewers and recommend accept. A remaining concern is that the proof of the only theoretical result, Theorem 1, is very similar to ref [Yang 2022b]. The authors addressed this concern in the response. The authors should clearly identify this similarity and discuss the comparisons and differences in their final version.

**Justification For Why Not Higher Score:**

The proof similarity to an existing study [Yang 2022b] should be clarified.

**Justification For Why Not Lower Score:**

The idea is interesting and the proposed method is technically sould and well verified.

---

### Decision · Program_Chairs · 2024-01-16

Accept (poster)